# Attacking Perceptual Similarity Metrics

**Abhijay Ghildyal**                                                                  *abhijay@pdx.edu*
*Department of Computer Science*
*Portland State University*

**Feng Liu**                                                                          *fliu@pdx.edu*
*Department of Computer Science*
*Portland State University*

**Reviewed on OpenReview:** *https://openreview.net/forum?id=r9vGSpbbRO*

## Abstract

Perceptual similarity metrics have progressively become more correlated with human judgments on perceptual similarity; however, despite recent advances, the addition of an imperceptible distortion can still compromise these metrics. In our study, we systematically examine the robustness of these metrics to imperceptible adversarial perturbations. Following the two-alternative forced-choice experimental design with two distorted images and one reference image, we perturb the distorted image closer to the reference via an adversarial attack until the metric flips its judgment. We first show that all metrics in our study are susceptible to perturbations generated via common adversarial attacks such as FGSM, PGD, and the One-pixel attack. Next, we attack the widely adopted LPIPS metric using spatial-transformation-based adversarial perturbations (stAdv) in a white-box setting to craft adversarial examples that can effectively transfer to other similarity metrics in a black-box setting. We also combine the spatial attack stAdv with PGD ($\ell_\infty$-bounded) attack to increase transferability and use these adversarial examples to benchmark the robustness of both traditional and recently developed metrics. Our benchmark provides a good starting point for discussion and further research on the robustness of metrics to imperceptible adversarial perturbations. Code is available at https://tinyurl.com/attackingpsm.

## 1 Introduction

Comparison of images using a similarity measure is crucial for defining the quality of an image for many applications in image and video processing. Recently, perceptual similarity metrics have become vital for optimizing and evaluating deep neural networks used in low-level computer vision tasks (Dosovitskiy & Brox, 2016; Zhu et al., 2016; Johnson et al., 2016; Ledig et al., 2016; Sajjadi et al., 2017; Kettunen et al., 2019a; Zhang et al., 2020; Son et al., 2020; Niklaus & Liu, 2020; Karras et al., 2020). Learned perceptual image patch similarity (LPIPS) metric by Zhang et al. (2018b) is one such widely adopted perceptual similarity metric. Apart from these image enhancement and generation tasks, similarity metrics are also used in optimizing, constraining, and evaluating adversarial attacks (Szegedy et al., 2014; Goodfellow et al., 2015; Carlini & Wagner, 2017; Kurakin et al., 2017; Hosseini & Poovendran, 2018; Dong et al., 2018; Shamsabadi et al., 2020; Laidlaw & Feizi, 2019). A limitation in early adversarial robustness studies has been the use of $\ell_p$ norms as a distance metric to judge the imperceptibility of synthesized adversarial perturbations. These attack methods optimized for stronger adversarial perturbations while keeping the perturbations within imperceptibility levels via an $\ell_p$ norm. However, as we now know, $\ell_p$ distance metrics are not a good proxy to human perception, and several learned perceptual similarity metrics have been developed to correlate better with human judgment. More recently, Laidlaw et al. (2020) proposed neural perceptual threat models (NPTM) and subsequently a defense method that could generalize well against unforeseen adversarial attacks, in which, instead of an $\ell_p$ norm, the severity, or perceptibility of the adversarial perturbations, is bounded by LPIPS, a learned perceptual similarity metric. Hence, they employed LPIPS in their optimization to

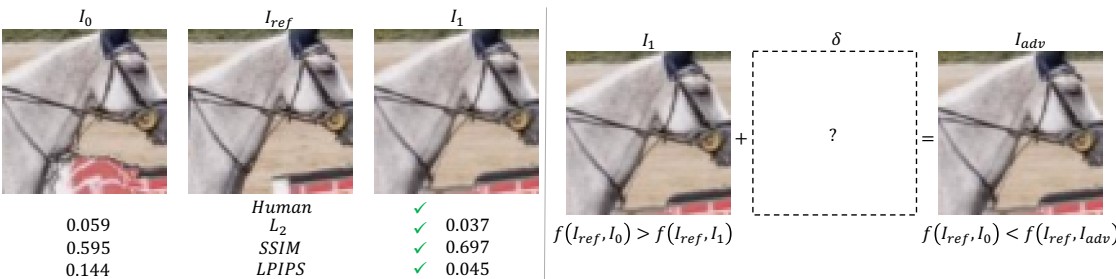

Figure 1: $I_1$ is more similar to $I_{ref}$ than $I_0$ according to all perceptual similarity metrics and humans. We attack $I_1$ by adding imperceptible adversarial perturbations ($\delta$) such that the metric ($f$) flips its earlier assigned rank, i.e., in the above sample, $I_0$ becomes more similar to $I_{ref}$.

generate adversarial examples. However, it remains unanswered whether LPIPS itself is robust towards imperceptible adversarial perturbations. The question then arises, *"How robust are perceptual similarity metrics against imperceptible adversarial perturbations?"* We posit that more accurate and robust perceptual similarity metrics can lead to stronger defenses against adversarial threats. In a recent study, Mahloujifar et al. (2020) showed that a better perception model to test the imperceptibility of adversarial perturbations can lead to stronger robustness guarantees for image classifiers.

We begin by examining whether it is possible to find imperceptible adversarial perturbations that can overturn perceptual similarity judgments. It is well known that machine learning models are easy to fool with adversarial perturbations imperceptible to the human eye (Szegedy et al., 2014). Interestingly, similar imperceptible perturbations can bring about a sizeable change in the measured distance of a distorted image from its reference. As shown in Figure 1, we examine this change in measured distances using a two-alternative forced choice (2AFC) test example, where the participants were asked, "which of the two distorted images ($I_0$ and $I_1$) is more similar to the reference image ($I_{ref}$)". Then, we apply an imperceptible perturbation to the distorted image that has the lower perceptual distance (i.e., more similar to $I_{ref}$) to see if the similarity judgment for the sample overturns. In such a scenario, human opinion remains the same, while perceptual similarity metrics often overturn their judgment. Perceptual similarity metrics measure the similarity between two images and are widely used in many real-world applications. Having a robust metric is sometimes critical. Copyright protection is one critical use case where automatic image similarity assessment plays an important role. A malicious user can upload copyright-protected images with imperceptible perturbations, making the images less detectable on the internet. Interestingly, recent work began to investigate the perceptual robustness of image quality assessment (IQA) methods via adversarial perturbations Zhang et al. (2022) and Lu et al. (2022). However, these studies focus on no-reference image quality assessment methods. The robustness of perceptual similarity metrics, often used as full-reference image quality assessment methods, has been less studied.

There are two popular approaches to examining the robustness of perceptual similarity metrics: (1) addition of small amounts of hand-crafted distortions such as translation, rotation, dilation, JPEG compression, and Gaussian blur, and (2) analysis of more advanced adversarial perturbations. For the former, seminal contributions have been made (Ma et al., 2018; Ding et al., 2020; Bhardwaj et al., 2020; Gu et al., 2020). However, in contrast to previous work, we focus on performing the latter as it has not received considerable attention. In our work, we demonstrate that threats to similarity metrics can be easily created using common gradient-based iterative white-box attacks, such as fast gradient sign method (FGSM) (Goodfellow et al., 2015) and projected gradient descent (PGD) (Madry et al., 2018). These attacks do not deform the structure but rather manipulate pixel values in the image. In recent research, questions regarding the robustness of perceptual similarity metrics towards geometric distortions are of central interest (as discussed above). Hence, we also use the spatial adversarial attack stAdv (Xiao et al., 2018), which geometrically deforms the image. It utilizes optical flow for crafting perturbations in the spatial domain. We use this attack to generate adversarial samples for comparing the robustness of various metrics.

We also examine whether perceptual metrics can be attacked in black box settings. To this end, we first use the One-pixel attack (Su et al., 2019) that uses differential evolution (Storn & Price, 1997) to optimize a single-pixel perturbation on the adversarial image. While compared to white box attacks such as FGSM and

PGD, this One-pixel attack does not need the model parameters of a similarity metric, it needs to access its output. Therefore, we furthermore explore transferable attacks (Liu et al., 2017; Xie et al., 2018; 2019) which requires no information about the model. Specifically, we generate adversarial examples using the parameters of a source model and use them to attack a target model. In our study, we use LPIPS(AlexNet) as the source model and attack it via stAdv. We extend the successfully attacked examples onto a target perceptual similarity metric. It is a black-box setting as it does not require access to the target perceptual metric's parameters. In our work, we combine stAdv (spatial attack) with PGD ($\ell_\infty$-bounded attack) that strengthens the severity of the adversarial examples.

The main contribution of this paper is the first systematical investigation on whether existing perceptual similarity metrics are susceptible to state-of-the-art adversarial attacks. Our study includes a set of carefully selected attacking methods and a wide variety of perceptual similarity metrics. Our study shows that all these similarity metrics, including both traditional quality metrics and recent deep learning-based metrics, can be successfully attacked by both white-box and black-box attacks.

## 2 Related Work

Earlier metrics such as SSIM (Wang et al., 2004) and FSIMc (Zhang et al., 2011) were designed to approximate the human visual systems' ability to perceive and distinguish images, specifically using statistical features of local regions in the images. Whereas, recent metrics (Zhang et al., 2018b; Prashnani et al., 2018; Ma et al., 2018; Kettunen et al., 2019b; Ding et al., 2020; Bhardwaj et al., 2020; Ghildyal & Liu, 2022) are deep neural network based approaches that learn from human judgments on perceptual similarity. LPIPS (Zhang et al., 2018b) is one such widely used metric. It leverages the activations of a feature extraction network at each convolutional layer to compute differences between two images which are then passed on to linear layers to finally predict the perceptual similarity score. Prashnani et al. (2018) developed the Perceptual Image Error Metric (PieAPP) that uses a weight-shared feature extractor on each of the input images, followed by two fully-connected networks that use the difference of those features to generate patch-wise errors and corresponding weights. The weighted average of the errors is the final score. Liu et al. (2022) used the Swin Transformer (Liu et al., 2021) for multi-scale feature extraction in their metric, Swin-IQA. Its final score is the average across all cross-attention operations on the difference between the features. Swin-IQA performs better than the CNN-based metrics in accurately ranking, according to human opinion, the distorted images synthesized by methods from the Challenge on Learned Image Compression (CLIC, 2022).

In recent years, apart from making the perceptual similarity metrics correlate well with human opinion, there has been growing interest in examining their robustness towards geometric distortions. Wang & Simoncelli (2005) noted that geometric distortions cause consistent phase changes in the local wavelet coefficients while structural content stays intact. Accordingly, they developed complex wavelet SSIM (CW-SSIM) that used phase correlations instead of spatial correlations, making it less sensitive to geometric distortions. Ma et al. (2018) benchmarked the sensitivity of various metrics against misalignment, scaling artifacts, blurring, and JPEG compression. They then trained a CNN with augmented images to create the geometric transformation invariant metric (GTI-CNN). In a similar study, Ding et al. (2020) suggested computing global measures instead of pixel-wise differences and then blurred the feature embeddings by replacing the max pooling layers with $l_2$-pooling layers. It made their metric, deep image structure and texture similarity (DISTS), robust to local and global distortions. Ding et al. (2021) extend DISTS making it robust for perceptual optimization of image super-resolution methods. They separate texture from the structure in the extracted multi-scale feature maps via a dispersion index. Then, to compute feature differences for the final similarity score, they modify SSIM by adaptively weighting its structure and texture measurements using the dispersion index. Bhardwaj et al. (2020) developed the perceptual information metric (PIM). PIM has a pyramid architecture with convolutional layers that generate multi-scale representations, which get processed by dense layers to predict mean vectors for each spatial location and scale. The final score is estimated using symmetrized KL divergence using Monte Carlo sampling. PIM is well correlated with human opinions and is robust against small image shifts, even though it is just trained on consecutive frames of a video, without any human judgments on perceptual similarity. Czolbe et al. (2020) used Watson's perceptual model (Watson, 1993) and replaced discrete cosine transform with discrete fourier transform (DFT) to develop a perceptual similarity loss function robust against small shifts. Kettunen et al. (2019b) compute the average LPIPS

score over an ensemble of randomly transformed images. Their self-ensembling metric E-LPIPS is robust to the Expectations over Transformations attacks (Athalye et al., 2018; Carlini & Wagner, 2017). Our attack approach is similar to an attack investigated by Kettunen et al. (2019b), where the adversarial images look similar but have a large LPIPS distance (smaller distance means more similarity). However, they only investigate the LPIPS metric. Ghildyal & Liu (2022) develop a shift-tolerant perceptual metric that is robust to imperceptible misalignments between the reference and the distorted image. For it, they test various neural network elements and modify the architecture of the LPIPS metric rather than training it on augmented data to handle the misalignment, making it more consistent with human perception. So far, the majority of prior research has focused on geometric distortions, while no study has systematically investigated the robustness of various similarity metrics to more advanced adversarial perturbations that are more perceptually indistinguishable. We seek to address this critical open question, *whether perceptual similarity metrics are robust against imperceptible adversarial perturbations.* In our paper, we show that the metrics often overturn their similarity judgment after the addition of adversarial perturbations, unlike humans, to whom the perturbations are unnoticeable.

There exists a considerable body of literature on adversarial attacks (Szegedy et al., 2014; Goodfellow et al., 2015; Liu et al., 2017; Papernot et al., 2016; Carlini & Wagner, 2017; Xie et al., 2018; Hosseini & Poovendran, 2018; Madry et al., 2018; Xiao et al., 2018; Brendel et al., 2018; Song et al., 2018; Zhang et al., 2018a; Engstrom et al., 2019; Laidlaw & Feizi, 2019; Su et al., 2019; Wong et al., 2019; Bhattad et al., 2019; Xie et al., 2019; Zeng et al., 2019; Dolatabadi et al., 2020; Tramèr et al., 2020; Laidlaw et al., 2020; Croce et al., 2020; Wu & Zhu, 2020), but none of the previous investigations have ever considered attacking perceptual similarity metrics, except for E-LPIPS (Kettunen et al., 2019b) which only studies the LPIPS metric. This paper builds upon this line of research and carefully selects a set of representative attacking algorithms to investigate the adversarial robustness of similarity metrics. We briefly describe these methods and how we employ them to attack similarity metrics in Section 3. In parallel, Lu et al. (2022) developed an adversarial attack for neural image assessment (NIMA) (Talebi & Milanfar, 2018) to prevent misuse of high-quality images on the internet. NIMA is NR-IQA, while we systematically investigate several FR-IQA methods against various attacks.

Recent work underlines the importance of perceptual distance as a bound for adversarial attacks (Laidlaw et al., 2020; Wang et al., 2021; Zhang et al., 2022). Laidlaw et al. (2020) developed a neural perceptual threat model (NPTM) that employs the perceptual similarity metric LPIPS(AlexNet) as a bound for generating adversarial examples and provided evidence that $l_p$-bounded and spatial attacks are near subsets of the NPTM. Similarly, Zhang et al. (2022) developed a perceptual threat model to attack no-reference IQA methods by constraining the perturbations via full-reference IQA, i.e., perceptual similarity metrics such as SSIM, LPIPS, and DISTS. They posit that the metrics are "approximations to human perception of just-noticeable differences" (Zhang et al., 2022), therefore, can keep perturbations imperceptible. Moreover, Laidlaw et al. (2020) found LPIPS to correlate well with human opinion when evaluating adversarial examples. *However, it has not yet been established whether LPIPS and other perceptual similarity metrics are adversarially robust.* We investigate this in our work, and the findings in our study indicate that all metrics, including LPIPS, are not robust to various kinds of adversarial perturbations.

## 3 Method

**Dataset.** Our study uses the Berkeley-Adobe perceptual patch similarity (BAPPS) dataset, originally used to train a perceptual similarity metric (Zhang et al., 2018b). Each sample in this dataset contains a set of 3 images: 2 distorted ($I_0$ and $I_1$) and 1 reference ($I_{ref}$). For perceptual similarity assessment, the scores were generated using a two-alternative forced choice (2AFC) test where the participants were asked, "which of two distortions is more similar to a reference" (Zhang et al., 2018b). For the validation set, 5 responses per sample were collected. The final human judgment is the average of the responses. The types of distortions in this dataset are traditional, CNN-based, and distortions by real algorithms such as super resolution, frame interpolation, deblurring, and colorization. Human opinions could be divided, i.e., all responses in a sample may not have voted for the same distorted image. In our study, to ensure that the two distorted images in the sample have enough disparity between them, we only select those samples where humans unanimously voted for one of the distorted images. In total, there are 12,227 such samples.

It is non-trivial to compare metrics based on a norm-based constraint simply because a change of 10% in metric A's score is not equal to a 10% change in metric B's score. But how does one calculate the fooling rate that measures the susceptibility of a similarity metric? A straightforward method is to compare all metrics against human perceptual judgment. The 2AFC test gathers human judgment on which of the two distorted images is more similar to the reference. Using this knowledge, we can benchmark various metrics and test whether their accuracy drops or, i.e. if they flip their judgment when attacked. To make it a fair challenge, we only use samples where human opinion completely prefers one distorted image over the other.

**Attack Models.** As observed in Figure 1, the addition of adversarial perturbations can lead to a rank flip. We make use of existing attack methods such as FGSM (Goodfellow et al., 2015), PGD (Madry et al., 2018), One-pixel attack (Su et al., 2019), and spatial attack stAdv (Xiao et al., 2018) to generate such adversarial samples. These attack methods were originally devised to fool image classification models, therefore, we introduce minor modifications in their procedures to attack perceptual similarity metrics. We select one of the distorted images, $I_0$ or $I_1$, that is more similar to $I_{ref}$ to attack. The distorted image being attacked is $I_{prey}$, and the other image is $I_{other}$; accordingly, for the sample in Figure 1, $I_1$ is $I_{prey}$ and $I_0$ is $I_{other}$. Consider $s_i$ as the similarity score between $I_i$ and $I_{ref}$[1]. Before the attack, the original rank is $s_{other} > s_{prey}$, but after the attack $I_{prey}$ turns into $I_{adv}$, and when the rank flips, $s_{adv} > s_{other}$. In image classification, a misclassification is used to measure the attack's success, while for perceptual similarity metrics, an attack is successful when the rank flips.

**Fast Gradient Sign Method.** FGSM is a popular white-box attack introduced by Goodfellow et al. (2015). This attack method projects the input image $I$ onto the boundary of an $\epsilon$ sized $\ell_\infty$-ball, and therefore, restricts the perturbations to the locality of $I$. We follow this method to generate imperceptible perturbations by constraining $\epsilon$ to be small for our experiments. This attack starts by first computing the gradient with respect to the loss function of the image classifier being attacked. The signed value of this gradient multiplied by $\epsilon$ generates the perturbation, and thus, $I_{adv} := I + \epsilon \cdot sign(\nabla_I J(\theta, I, target))$, where $\theta$ are the model parameters. We adopt this method to attack perceptual similarity metrics.

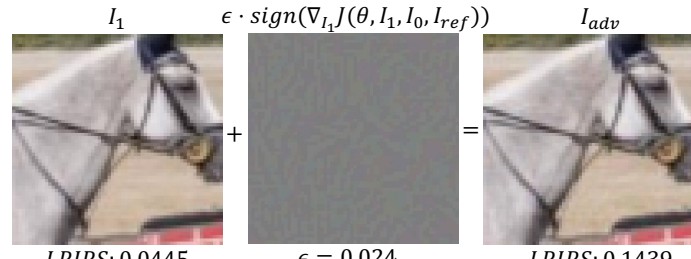

$$I_1 \qquad \epsilon \cdot sign(\nabla_{I_1} J(\theta, I_1, I_0, I_{ref})) \qquad I_{adv}$$

*LPIPS*: 0.0445 $\qquad \epsilon = 0.024 \qquad$ *LPIPS*: 0.1439

Figure 2: FGSM attack on LPIPS(Alex). In this white-box attack, we use the LPIPS network parameters to compute the signed gradient. With increase in $\epsilon$, the severity of the attack increases. In this example, the adversarial perturbations are hardly visible. The RMSE between the prey image $I_1$ and the adversarial image $I_{adv}$ is 3.53.

We formulate a new loss function for an untargeted attack as:

$$J(\theta, I_{prey}, I_{other}, I_{ref}) = \left( \frac{s_{other}}{s_{other} + s_{prey}} - 1 \right)^2 \tag{1}$$

We maximize this loss, i.e., move in the opposite direction of the optimization by adding the perturbation to the image. The human score of all the samples in our selected dataset is either 0 or 1, unanimous vote. Hence, we can easily employ the loss function in Equation 1, because if the metric predicts the rank correctly then $(s_{other}/(s_{other} + s_{prey}))$ would be $\approx 1$. Afterwards, if the attack is successful then $(s_{other}/(s_{other} + s_{adv}))$ becomes less than 0.5, causing the rank to flip. Algorithm 3 (refer Appendix B) provides the details for the FGSM attack. First, $I_{prey}$ is selected based on the original rank. The model parameters remain constant, and we compute the gradients with respect to the input image $I_{prey}$. To increase perturbations in normalized images, we increase the $\epsilon$ in steps of 0.0001 starting from 0.0001. When $\epsilon$ is large enough, the rank flips. It would mean that the attack was successful (see example in Figure 2). If the final value of $\epsilon$ is small then the perturbation is imperceptible, making it hard to discern any difference between the original image and its adversarial sample.

**Projected Gradient Descent.** PGD attack by Madry et al. (2018) takes a similar approach to FGSM, but instead of a single large step like in FGSM, PGD takes multiple small steps for generating perturbation

---

[1]smaller $s_i$ means $I_i$ is more similar to $I_{ref}$

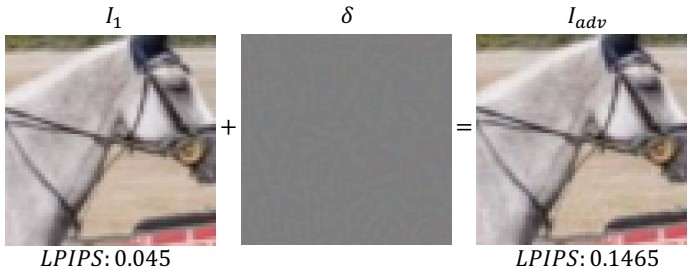

$I_1$         $\delta$         $I_{adv}$

$LPIPS: 0.045$         $LPIPS: 0.1465$

Figure 3: PGD attack on LPIPS(Alex). In this white-box attack, we use the LPIPS network parameters to compute the signed gradient. With increase in the number of attack iterations, the severity of the attack increases. In this example, perturbations in $I_{adv}$ are not visible. The RMSE between the prey image $I_1$ and the adversarial image $I_{adv}$ is 2.10.

$\delta$. Hence, the projection of $I$ stays either inside or on the boundary of the $\epsilon$-ball. This multistep attack is defined as:

$$I_{adv}^{t+1} = P_c\big(I_{adv}^t + \alpha \cdot sign(\nabla_{I_{adv}^t} J(\theta, I_{adv}^t, I_{other}, I_{ref}))\big) \tag{2}$$

where $J$ is the loss defined in Equation 1. The perturbation on each pixel is bounded to a predefined range using the projection constraint $P_c$. We implement $P_c$ using a clip operation on the final perturbation $\delta$ (Line 14 Algorithm 1). As shown in Algorithm 1, the signed gradient is multiplied with step size $\alpha$, and this adversarial perturbation is added to $I_{adv}^t$. The final perturbation $\delta$ is the difference between $I_{adv}^t$ and $I_{prey}$, and in our method, $\delta$ is bounded by $\ell_\infty$ norm. Hence, the PGD attack is an **$\ell_\infty$-bounded attack**.

**One-pixel Attack.** The previous two approaches are white-box attacks. We now use a black-box attack, the One-pixel attack by Su et al. (2019) that perturbs only a single pixel using differential evolution (Storn & Price, 1997).

The objective of the One-pixel attack is defined as:

---

**Algorithm 1:** PGD attack on Similarity Metrics

**Input:** $I_0, I_1, I_{ref}$, metric $f$, $\epsilon$ (perturbation limit 0.03
1  ), $max\_iterations$ (30), $\alpha$ (step size 0.001)
     **Output:** $attack\_success$ True on rank flip
2  $s_0 = f(I_{ref}, I_0)$; $s_1 = f(I_{ref}, I_1)$; $rank = int(s_0 > s_1)$;
3  // If $I_0$ is more similar to $I_{ref}$ then $rank$ is 0 else 1
4  **if** $rank = 1$ **then** $I_{prey} = I_1$; $s_{other} = s_0$;
5  **else** $I_{prey} = I_0$; $s_{other} = s_1$;
6  $\delta = zeros\_like(I_{prey})$ // perturbation
7  $k = 0$
8  **while** $k \leq max\_iterations$ **do**
9     $I_{adv} = clip(I_{prey} + \delta, min = -1, max = 1)$
10    $s_{adv} = f(I_{ref}, I_{adv})$
11    **if** $s_{adv} > s_{other}$ **then return** *True* // Attack successful
12    $J = \big((s_{other}/(s_{other} + s_{adv})) - 1\big)^2$ // Loss
13    $signed\_grad = sign\big(\nabla_{I_{adv}} J\big)$
14    $I'_{adv} = I_{adv} + \alpha * signed\_grad$
15    $\delta = clip(I'_{adv} - I_{prey}, min = -\epsilon, max = +\epsilon)$
16    $k = k + 1$
17 **return** *False* // Attack unsuccessful

---

$$\begin{aligned} \underset{e(I_{prey})^*}{\text{maximize}} \quad & f(I_{prey} + e(I_{prey}), I_{ref}) \\ \text{subject to} \quad & ||e(I_{prey})||_0 \leq d \end{aligned} \tag{3}$$

where $f$ is the similarity metric, and the vector $e(I_{prey})$ is the additive adversarial perturbation, and $d$ is 1 for the One-pixel attack. This algorithm aims to find a mutation to one particular pixel such that a similarity metric $f$, such as LPIPS, will consider $I_{prey}$ is less similar to $I_{ref}$ than it is originally, and thus, the rank is flipped. Note, for LPIPS, a larger score indicates the two images being less similar. Please refer to Su et al. (2019) for more details of this attack algorithm. For attack example, please refer to Figure 8 in Appendix C.

**Spatial Attack (stAdv).** The goal of the stAdv attack is to deform the image geometrically by displacing pixels (Xiao et al., 2018). It generates adversarial perturbations in the spatial domain rather than directly manipulating pixel intensity values. This attack synthesizes the spatially distorted adversarial image ($I_{adv}$) via optimizing a flow vector and backward warping with the input image ($I_{prey}$) using differentiable bilinear interpolation (Jaderberg et al., 2015). For each sample, we start with a flow initialized with zeros and then optimize it using L-BFGS (Liu & Nocedal, 1989) for the following loss.

$$\mathcal{L} = \alpha \mathcal{L}_{rank} + \beta \mathcal{L}_{flow} \tag{4}$$

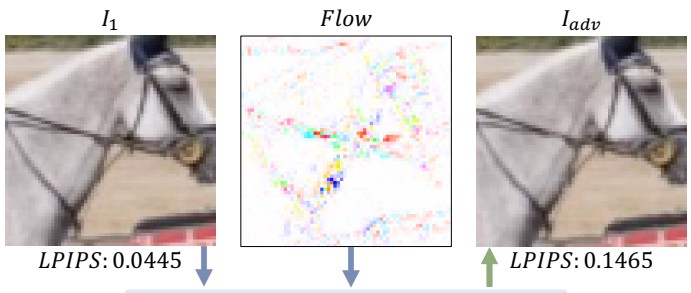

Figure 4: Spatial attack stAdv on LPIPS(AlexNet). We attack LPIPS(AlexNet) to create adversarial images. This attack optimizes a flow vector to create perturbations in the spatial domain. In this example, flow distorts the structure of the horse to generate the adversarial image. The RMSE between the prey image $I_1$ and the adversarial image $I_{adv}$ is 2.50.

$$\mathcal{L}_{flow} = \sum_p^{pixels} \sum_q^{neighbors(p)} \sqrt{(u_p - u_q)^2 + (v_p - v_q)^2} \tag{5}$$

where $(u, v)$ is the displacement vector at pixel location $p$ and its 4 neighbors $q$.

$$\mathcal{L}_{rank} = \left(\frac{s_{other}}{s_{other} + s_{adv}}\right)^2 \tag{6}$$

where $\alpha$ is 50 and $\beta$ is 0.05.

As we minimize $\mathcal{L}_{rank}$, the perturbations in $I_{adv}$ will increase, and thus rank will flip. Simultaneously, we also minimize $\mathcal{L}_{flow}$ which defines the amount of perturbations generated by flow to distort the image. It enforces the perturbations to be constrained to make as little change to the attacked image $I_{prey}$ as possible. Xiao et al. (2018) performed a user study to test the perceptual quality of the images having perturbations generated by the stAdv attack and found them to be indistinguishable by humans. By visual inspection, we found the adversarial perturbations on the images imperceptible in our studies as well.

---

**Algorithm 2:** stAdv attack on LPIPS

**Input:** $I_0$, $I_1$, $I_{ref}$, LPIPS $f$, *max_iterations* (250)
**Output:** *attack_success* True on rank flip
1 **Function** stAdv_attack(*flow*, $f$, $I_{prey}$, $I_{ref}$, $s_{other}$):
2     $I_{adv} = \text{warp}(flow, I_{prey})$ // Backwarp via bilinear interpolation
3     $s_{adv} = f(I_{ref}, I_{adv})$
4     $\mathcal{L}_{rank}, \mathcal{L}_{perturb} = calc\_loss(I_{ref}, I_{prey}, I_{adv}, s_{other}, f)$
5     $\mathcal{L} = \mathcal{L}_{rank} + \mathcal{L}_{perturb}$
6     $gradient = \nabla_{flow}\mathcal{L}$
7     **if** $s_{adv} > s_{other}$ **then return** *0, gradient, flow* // Attack successful
8     **else return** $\mathcal{L}$, *gradient, flow* // Attack unsuccessful

1 $s_0 = f(I_{ref}, I_0)$; $s_1 = f(I_{ref}, I_1)$;
2 $rank = int(s_0 > s_1)$ // If $I_0$ is more similar to $I_{ref}$ then *rank* is 0 else 1
3 **if** $rank = 1$ **then** $I_{prey} = I_1$; $s_{other} = s_0$;
4 **else** $I_{prey} = I_0$; $s_{other} = s_1$;
5 // Initialize a flow vector with zeros
6 $flow = zeros\_like(2 * I_{prey}$ height $* I_{prey}$ width$)$
7 *converge, grad, flow* = $L\text{-}BFGS$(func=$stAdv\_attack$, args=(*flow*, $f$, $I_{prey}$, $I_{ref}$, $s_{other}$), iterations=*max_iterations*) // Optimize flow vector
8 **if** $converge = 0$ **then** *attack_success* = True
9 **else** *attack_success* = False
10 **return** *attack_success*

---

## 4 Experiments and Results

We experiment with a wide variety of similarity metrics including both traditional ones, such as L2, SSIM (Wang et al., 2004), MS-SSIM (Wang et al., 2003), CW-SSIM (Wang & Simoncelli, 2005) and FSIMc (Zhang et al., 2011), and the recent deep learning based ones, such as WaDIQaM-FR (Bosse et al., 2018), GTI-CNN (Ma et al., 2018), LPIPS (Zhang et al., 2018b), E-LPIPS (Kettunen et al., 2019b), DISTS (Ding et al., 2020), Watson-DFT (Czolbe et al., 2020), PIM (Bhardwaj et al., 2020), A-DISTS (Ding et al., 2021), ST-LPIPS (Ghildyal & Liu, 2022), and Swin-IQA (Liu et al., 2022). We adopt the BAPPS validation dataset (Zhang et al., 2018b) for our experiments. Following Zhang et al. (2018b) we scale the image

patches from size $256 \times 256$ to $64 \times 64$. As mentioned in Section 3, we believe that the predicted rank by a metric will be easy to flip on samples close to the decision boundary; therefore, we take a subset of the samples in the dataset which have a clear winner, i.e., all human responses indicated that one was distinctly better than the other. Now, in our dataset, we have 12,227 samples. We report the accuracy of metrics on the subset of selected samples and compare it with their

Table 1: Accuracy on the subset selected for our experiments correlates with the 2AFC score computed on the complete BAPPS validation dataset.

| Network | 2AFC (%) on complete BAPPS (36344 samples) | Accuracy (%) on subset of BAPPS (12227 samples) |
|---|---|---|
| L2 | 63.2 | 79.7 |
| SSIM (Wang et al., 2004) | 63.1 | 80.8 |
| WaDIQaM-FR (Bosse et al., 2018) | 66.5 | 83.3 |
| LPIPS(Alex) (Zhang et al., 2018b) | 69.8 | 92.4 |
| LPIPS(VGG) (Zhang et al., 2018b) | 68.1 | 89.8 |
| DISTS (Ding et al., 2020) | 68.9 | 91.3 |

Two-alternative forced choice (2AFC) scores on the complete BAPPS validation dataset. As shown in Table 1, all these metrics consistently correlated better with the human opinions on the subset of BAPPS than on the full dataset, which is expected as we removed the ambiguous cases.

In this section, we first show that similarity metrics are susceptible to both white-box and black-box attacks. Based on this premise, we hypothesize that these similarity metrics are vulnerable to transferable attacks. To prove this, we attack the widely adopted LPIPS using the spatial attack stAdv to create adversarial examples and use them to benchmark the adversarial robustness of these similarity metrics. Furthermore, we add a few iterations of the PGD attack, hence combining our spatial attack with $\ell_\infty$-bounded perturbations, to enhance transferability to other perceptual similarity metrics.

## 4.1 Adversarial Attack on Perceptual Similarity Metrics

Through the following study, we test our hypothesis that similarity metrics are susceptible to adversarial attacks. We first determine whether it is possible to create imperceptible adversarial perturbations that can overturn the perceptual similarity judgment, i.e., flip the rank of the images in the sample. We try to achieve this by simply attacking with widely used white-box attacks like FGSM, and PGD, and a black-box attack like the One-pixel attack. As reported in Table 2, all these attacks can successfully flip the rank assigned by both traditional metrics such as L2, and SSIM (Wang et al., 2004), and learned metrics such as WaDIQaM-FR (Bosse et al., 2018), LPIPS (Zhang et al., 2018b), and DISTS (Ding et al., 2020), in a significant amount of samples.

For the PGD attack, the maximum $\ell_\infty$-norm perturbation[2] cannot be more than 0.03 as the step size $\alpha$ is 0.001, and the maximum attack iterations is 30. We chose 30 after visually inspecting for the imperceptibility of perturbations on the generated adversarial samples. With the same threshold, the FGSM attack would not be as successful as PGD, which we show in Appendix E. Therefore, to report the results of the FGSM attack, based on empirical evaluation, we select the maximum $\epsilon$ as 0.05. We present the results separately for samples where the originally predicted rank by the metric matches the rank provided by humans. Now, focusing only on the samples where the metric matches with the ranking by humans, we found L2 and DISTS to be the most robust against FGSM and PGD with only about 30% of the samples flipped, while LPIPS and WadIQaM-FR were the least robust, with about 80% of the samples flipped. The same conclusion can also be reached by observing $\epsilon$ (or perturbations) required to attack them. Next, despite being a black-box attack, the One-pixel attack can also successful flip ranks. LPIPS(AlexNet) has the least robustness to the One-pixel attack with 82% of the samples flipped, and this lack of adversarial robustness is consistent across all three attacks. SSIM and WadIQaM-FR are more robust to this attack, with only 18% and 31% samples flipped. It is interesting to note that similar results are achievable by using just the score of the adversarial image, i.e., $s_{adv}$ as loss for optimization.

Not surprisingly, it is easier to flip rank for the samples where the metric does not match with human opinion. As reported in Table 2, a much higher number of those samples flip where the rank by metric and humans did not match. These samples have a lower $\epsilon$, which means that lesser perturbations were required to flip the rank. We attribute the easy rank-flipping for these samples to the fact that the distorted images in each sample, i.e., $I_{other}$ and $I_{prey}$, are much closer to the decision boundary for the rank flip.

---

[2]All $\epsilon$ (or perturbation) in this paper were computed from normalized images in the range [-1,1].

Table 2: FGSM, PGD, and One-pixel attack results. Larger $\epsilon$ allows more perturbations, and lower RMSE relates to higher imperceptibility.

| Network | Same Rank by Human & Metric | Total Samples | FGSM ($\epsilon < 0.05$) | | | | PGD | | | | | | One-pixel |
| | | | #Samples Flipped | Mean $\epsilon$ | RMSE | | #Samples Flipped | % pixels with $\epsilon$ | | | RMSE | | #Samples Flipped |
| | | | | | $\mu$ | $\sigma$ | | >0.001 | >0.01 | >0.03 | $\mu$ | $\sigma$ | |
|---|---|---|---|---|---|---|---|---|---|---|---|---|---|
| L2 | ✓ | 9750 | 3759/39% | 0.023 | 2.9 | 1.7 | 2348/24% | 84.4 | 56.1 | 0.0 | 1.9 | 1.0 | 4225/43% |
| | ✗ | 2477 | 1550/63% | 0.017 | 2.2 | 1.6 | 1202/49% | 82.0 | 42.7 | 0.0 | 1.5 | 1.0 | 1412/57% |
| SSIM | ✓ | 9883 | 6922/70% | 0.018 | 2.5 | 1.7 | 5297/54% | 94.6 | 53.6 | 0.0 | 1.8 | 1.0 | 1787/18% |
| (Wang et al., 2004) | ✗ | 2344 | 2013/86% | 0.011 | 1.6 | 1.3 | 1843/79% | 87.3 | 32.0 | 0.0 | 1.3 | 0.8 | 1005/43% |
| WadIQaM-FR | ✓ | 10191 | 8841/87% | 0.006 | 1.0 | 1.0 | 10176/100% | 69.2 | 4.3 | 0.0 | 0.7 | 0.3 | 3130/31% |
| (Bosse et al., 2018) | ✗ | 2036 | 2012/100% | 0.001 | 0.6 | 0.3 | 2035/100% | 41.2 | 0.1 | 0.0 | 0.5 | 0.1 | 1598/79% |
| LPIPS(Alex) | ✓ | 11303 | 7247/64% | 0.018 | 2.4 | 1.7 | 8806/78% | 86.8 | 28.7 | 0.0 | 1.3 | 0.6 | 9255/82% |
| (Zhang et al., 2018b) | ✗ | 924 | 912/99% | 0.004 | 0.9 | 0.7 | 917/99% | 59.5 | 3.2 | 0.0 | 0.8 | 0.3 | 921/100% |
| LPIPS(VGG) | ✓ | 10976 | 8434/77% | 0.012 | 1.7 | 1.5 | 9689/88% | 81.6 | 15.6 | 0.0 | 1.0 | 0.5 | 7212/66% |
| (Zhang et al., 2018b) | ✗ | 1251 | 1244/100% | 0.003 | 0.8 | 0.5 | 1246/100% | 52.3 | 1.6 | 0.0 | 0.7 | 0.2 | 1219/98% |
| DISTS | ✓ | 11158 | 3043/27% | 0.025 | 3.3 | 1.8 | 2306/21% | 97.0 | 75.4 | 0.0 | 2.6 | 1.3 | 7416/67% |
| (Ding et al., 2020) | ✗ | 1069 | 795/74% | 0.016 | 2.2 | 1.7 | 723/68% | 91.9 | 50.0 | 0.0 | 2.0 | 1.3 | 1033/97% |

To test this, we calculate the absolute difference between $s_{other}$ and $s_{prey}$, i.e., the perceptual distances of $I_{other}$ and $I_{prey}$ from $I_{ref}$. As reported in Table 3, the similarity difference for these samples is much lesser than samples where the rank predicted by metric is the same as the rank assigned by humans. This result indicates that the samples where rank predicted by metric is not the same as the rank assigned by humans lie closer to the decision boundary, causing them to flip easier.

**Imperceptibility.** We discuss the imperceptibility of the adversarial perturbations by comparing the root mean square error (RMSE[3]) between the original and the perturbed image. As expected, the PGD attack is stronger than FGSM as it is capable of flipping a significant number of samples with lesser adversarial perturbations. In Appendix E, we experiment with increasing step size $\alpha$ for the PGD attack, which further increases its severity.

As reported in Table 2, for the PGD attack, a good portion of the adversarial image ($I_{adv}$) has $\epsilon < 0.01$, while for FGSM, the amount of pixel perturbation all over the image is a constant $\epsilon$ value which moreover is higher for a successful attack. Thus, on average, the $I_{adv}$ generated via PGD has lower RMSE and a higher PSNR (see Table 4) with the original image $I_{prey}$, compared to the $I_{adv}$ generated via FGSM. We also perform a visual sanity check and find the perturbations satisfactorily imperceptible. Only a single pixel is perturbed for $I_{adv}$ generated via the One-pixel attack, which we consider suitably imperceptible.

### 4.2 Transferable Adversarial Attack

In a real-world scenario, the attacker may not have access to the metric's architecture, hyper-parameters, data, or outputs. In such a scenario, a practical solution for the attacker is to transfer adversarial examples crafted on a source metric to a target perceptual similarity metric. Previous studies have suggested reliable approaches for creating such black-box transferable

Table 3: Comparing samples where the rank by metric was the same as assigned by humans versus samples where it was not.

| Network | Same Rank by Human & Metric | Similarity Diff. $abs(s_0 - s_1)$ |
|---|---|---|
| L2 | ✓ | 0.036 |
| | ✗ | 0.025 |
| SSIM | ✓ | 0.114 |
| (Wang et al., 2004) | ✗ | 0.054 |
| WadIQaM-FR | ✓ | 0.231 |
| (Bosse et al., 2018) | ✗ | 0.064 |
| LPIPS(Alex) | ✓ | 0.169 |
| (Zhang et al., 2018b) | ✗ | 0.024 |
| LPIPS(VGG) | ✓ | 0.174 |
| (Zhang et al., 2018b) | ✗ | 0.037 |
| DISTS | ✓ | 0.103 |
| (Ding et al., 2020) | ✗ | 0.022 |

Table 4: Comparing PSNR of adversarial images generated via FGSM vs. PGD. The $\epsilon$ for the adversarial images generated via FGSM is $< 0.05$. A higher mean PSNR of the PGD examples shows that the adversarial perturbations are less perceptible.

| Network | Same Rank by Human & Metric | FGSM | | PGD | |
| | | PSNR | | PSNR | |
| | | $\mu$ | $\sigma$ | $\mu$ | $\sigma$ |
|---|---|---|---|---|---|
| L2 | ✓ | 40.81 | 6.49 | 44.15 | 5.49 |
| | ✗ | 43.75 | 7.00 | 46.08 | 5.70 |
| SSIM | ✓ | 42.51 | 6.55 | 44.60 | 5.31 |
| (Wang et al., 2004) | ✗ | 46.39 | 6.09 | 47.19 | 5.16 |
| WadIQaM-FR | ✓ | 50.81 | 5.60 | 52.19 | 3.47 |
| (Bosse et al., 2018) | ✗ | 53.92 | 3.25 | 54.35 | 2.73 |
| LPIPS(Alex) | ✓ | 42.80 | 6.70 | 46.82 | 4.09 |
| (Zhang et al., 2018b) | ✗ | 49.98 | 4.19 | 50.80 | 3.14 |
| LPIPS(VGG) | ✓ | 45.96 | 6.38 | 48.68 | 3.72 |
| (Zhang et al., 2018b) | ✗ | 50.56 | 3.27 | 51.09 | 2.46 |
| DISTS | ✓ | 39.50 | 6.22 | 41.19 | 5.75 |
| (Ding et al., 2020) | ✗ | 43.64 | 6.95 | 44.41 | 6.39 |

---

[3]Throughout this paper, RMSE was calculated on images with pixel values ranging [0,255].

Table 5: Transferable adversarial attacks on perceptual similarity metrics. The adversarial examples were generated by attacking LPIPS(AlexNet) via stAdv. In total, there are 2726 samples. Next, we attacked LPIPS(AlexNet) using PGD(10). Then, we combined stAdv+PGD(10) by perturbing the stAdv generated images with PGD(10). Accurate samples are the ones for which the predicted rank by metric is equal to the rank assigned by humans. The transferability increases when the attacks are combined.

| Network | #Accurate Samples | # Accurate Samples Flipped | | | | | | |
|---|---|---|---|---|---|---|---|---|
| | | PGD(10) | PGD(20) | stAdv | stAdv + PGD(5) | stAdv + PGD(10) | stAdv + PGD(15) | stAdv + PGD(20) |
| L2 | 2099/77% | 101/5% | 174/8% | 77/4% | 134/6% | 189/9% | 200/10% | 257/12% |
| SSIM (Wang et al., 2004) | 2093/77% | 237/11% | 442/21% | 78/4% | 180/9% | 339/16% | 370/18% | 540/26% |
| MS-SSIM (Wang et al., 2003) | 2022/74% | 158/8% | 256/13% | 76/4% | 162/8% | 224/11% | 234/12% | 333/16% |
| CWSSIM (Wang & Simoncelli, 2005) | 1883/69% | 101/5% | 172/9% | 42/2% | 60/3% | 128/7% | 139/7% | 193/10% |
| FSIMc (Zhang et al., 2011) | 2025/74% | 222/11% | 325/16% | 202/10% | 233/12% | 302/15% | 310/15% | 393/19% |
| WaDIQaM-FR (Bosse et al., 2018) | 2083/76% | 95/5% | 186/9% | 59/3% | 85/4% | 146/7% | 156/7% | 238/11% |
| GTI-CNN (Ma et al., 2018) | 1946/71% | 448/23% | 480/25% | 494/25% | 488/25% | 504/26% | 510/26% | 543/28% |
| LPIPS(Squz.) (Zhang et al., 2018b) | 2503/92% | 298/12% | 656/26% | 114/5% | 221/9% | 519/21% | 555/22% | 886/35% |
| LPIPS(VGG) (Zhang et al., 2018b) | 2317/85% | 435/19% | 814/35% | 131/6% | 288/12% | 643/28% | 685/30% | 992/43% |
| E-LPIPS (Kettunen et al., 2019b) | 2442/90% | 503/21% | 643/26% | 517/21% | 552/23% | 641/26% | 655/27% | 817/33% |
| DISTS (Ding et al., 2020) | 2413/89% | 311/13% | 576/24% | 146/6% | 257/11% | 510/21% | 546/23% | 801/33% |
| Watson-DFT (Czolbe et al., 2020) | 2179/80% | 387/18% | 614/28% | 216/10% | 324/15% | 532/24% | 562/26% | 750/34% |
| PIM-1 (Bhardwaj et al., 2020) | 2468/91% | 696/28% | 814/33% | 756/31% | 772/31% | 826/33% | 852/35% | 958/39% |
| PIM-5 (Bhardwaj et al., 2020) | 2457/90% | 751/31% | 844/34% | 765/31% | 791/32% | 864/35% | 893/36% | 963/39% |
| A-DISTS (Ding et al., 2021) | 2346/86% | 339/14% | 661/28% | 164/7% | 276/12% | 561/24% | 590/25% | 850/36% |
| ST-LPIPS(Alex) (Ghildyal & Liu, 2022) | 2470/91% | 104/4% | 198/8% | 96/4% | 123/5% | 205/8% | 212/9% | 310/13% |
| ST-LPIPS(VGG) (Ghildyal & Liu, 2022) | 2493/91% | 210/8% | 453/18% | 103/4% | 153/6% | 321/13% | 360/14% | 576/23% |
| SwinIQA (Liu et al., 2022) | 2310/85% | 249/11% | 357/15% | 262/11% | 279/12% | 342/15% | 375/16% | 482/21% |

adversarial examples for image classifiers (Tramèr et al., 2017; Zhou et al., 2018; Inkawhich et al., 2019; Huang et al., 2019; Li et al., 2020; Hong et al., 2021). This paper focuses on perceptual similarity metrics and how they perform against such transferable adversarial examples. Specifically, we transfer the stAdv attack on LPIPS(AlexNet) to other metrics. We chose LPIPS(AlexNet) as it is widely adopted in many computer vision, graphics, and image / video processing applications. Furthermore, we combine the stAdv attack with PGD to increase the transferability of the adversarial examples to other metrics. In this study, we only consider samples for which the metrics and the human opinions agree on their rankings.

**stAdv.** As shown in Figure 4, stAdv has the capability of attacking high-level image features. As a white-box attack on LPIPS(AlexNet), out of the 11,303 accurate samples from total 12,227 samples, stAdv was able to flip judgment on 4658 samples with a mean RMSE of 2.37 with standard deviation 1.42. Because we need high imperceptibility, we remove samples with RMSE > 3 and are left with 3327 samples. We then perform a visual sanity check and remove some more with ambiguity, keeping only strictly imperceptible samples. In the end, we have 2726 samples, with a mean RMSE of 1.58 with standard deviation 0.63, which we transfer to other metrics as a black-box attack. As reported in Table 5, all metrics are prone to the attack. WaDIQaM-FR (Bosse et al., 2018) is most robust, while PIM (Bhardwaj et al., 2020) that was found robust to small imperceptible shifts is highly susceptible to this attack, although PIM is 15% more accurate than WaDIQaM-FR. DISTS, ST-LPIPS, and Swin-IQA have similar high accuracy as PIM but better robustness. Finally, we saw that, on average, learned metrics are more correlated with human opinions, but traditional metrics exhibit more robustness to the imperceptible transferable stAdv adversarial perturbations.

**PGD(10).** We now attack the original 2726 selected samples with the PGD attack. As shown in Section 4.1, perturbations generated via PGD have low perceptibility; hence, we create adversarial samples using PGD. In stAdv, we stopped the attack when the rank predicted by LPIPS(AlexNet) flipped. While in PGD, for comparison's sake, we fix the number of attack iterations to 10 for each sample to guarantee the transferability of perturbations. We call this transferable attack PGD(10), and the mean RMSE of the adversarial images generated is 1.28 with a standard deviation of 0.11. The metrics SSIM and WaDIQaM-FR are most robust to the transferable PGD(10) attack, as reported in Table 5.

**Combining stAdv and PGD(10).** The attacks stAdv and PGD are orthogonal approaches as PGD ($\ell_\infty$-bounded attack) manipulates the intensity of individual pixels while stAdv (spatial attack) manipulates the location of the pixels. We now combine the two by attacking the samples generated via stAdv with PGD(10). The mean RMSE of the generated adversarial images is 2.19 with a standard deviation of 0.41, just 0.61 higher than images generated via stAdv. As reported in Table 5, the increase in severity of the

adversarial perturbations in stAdv+PGD(10) leads to increased transferability. This result also is consistent with previous findings by Engstrom et al. (2019) where they combined PGD on top of their spatial attack and found that it leads to an additive increment in the misclassification rate.

**Summary.** In this paper, we successfully demonstrate that a wide variety of perceptual similarity metrics are susceptible to adversarial attacks. We show that adversarial perturbations crafted for LPIPS(AlexNet) generated via stAdv, can be transferred to other metrics. Furthermore, combining stAdv (spatial attack) with PGD ($\ell_\infty$-bounded attack) increases their transferability. We showcase a few examples in Figure 6 and Figure 7. In addition, the severity of the attack increases with the increasing number of PGD iterations (see Table 5). Our investigations also show that although more accurate, learned metrics may not be more robust than traditional ones (see Figure 5). Further tests carried out on two additional datasets and higher resolution images,

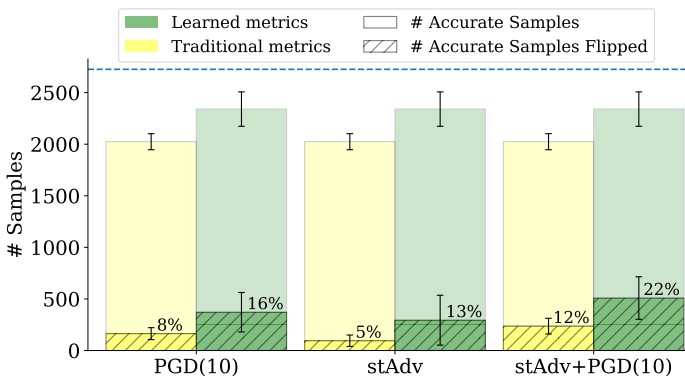

Figure 5: Comparing traditional metrics (L2, SSIM, MS-SSIM, CW-SSIM, and FSIMc) versus learned metrics (WaDIQaM-FR, GTI-CNN, LPIPS, DISTS, E-LPIPS, Watson-DFT, PIM, A-DISTS, ST-LPIPS, and Swin-IQA).

in Appendix D, corroborate with our previous results. We demonstrate the reverse of our attack in Appendix F, i.e., we attack the less similar of the two distorted images to make it more similar to the reference image. In summary, our findings point towards the need to develop robust perceptual similarity metrics.

## 5 Broader Impacts Statement

Perceptual similarity metrics have a wide variety of applications. Hence, there are benefits to studying the robustness of these metrics, and this work presents an opportunity to further improve the alignment of these metrics with human perception. At the same time, it is important to consider the negative outcomes of our work. Exposing the vulnerability of these metrics provides more details to malicious actors who would want to misuse this information to attack applications that make use of these similarity metrics in their pipeline, such as evading copyright detection. Perceptual similarity metrics can also be misused to synthesize malware images that could go undetected online. Therefore, we suggest further research on this topic to include appropriate defenses or more discussion on ways for mitigating such vulnerabilities. To aid further research on this topic, we shall make our code and data publicly available.

## 6 Conclusion

In this paper, we studied the robustness of various traditional and learned perceptual similarity metrics to imperceptible perturbations. We devised a methodology to craft such perturbations via adversarial attacks. Our findings suggest that, when comparing two images with respect to a reference, the addition of imperceptible distortions can overturn a metric's similarity judgment. The results of our study indicate that even learned perceptual metrics that match with human similarity judgments are susceptible to such imperceptible adversarial perturbations. We crafted adversarial examples using the spatial attack, stAdv, that were transferable to other metrics. We show that when combined with the PGD attack, the transferability of the adversarial examples can be further increased. Perceptual similarity metrics are designed to simulate the human visual system, and for this reason, these metrics are increasingly used in the assessment of image and video quality in real-world scenarios. Since invisible distortions can negatively impact the performance of similarity metrics, future studies for the design and development of newer metrics should also focus on validating robustness.

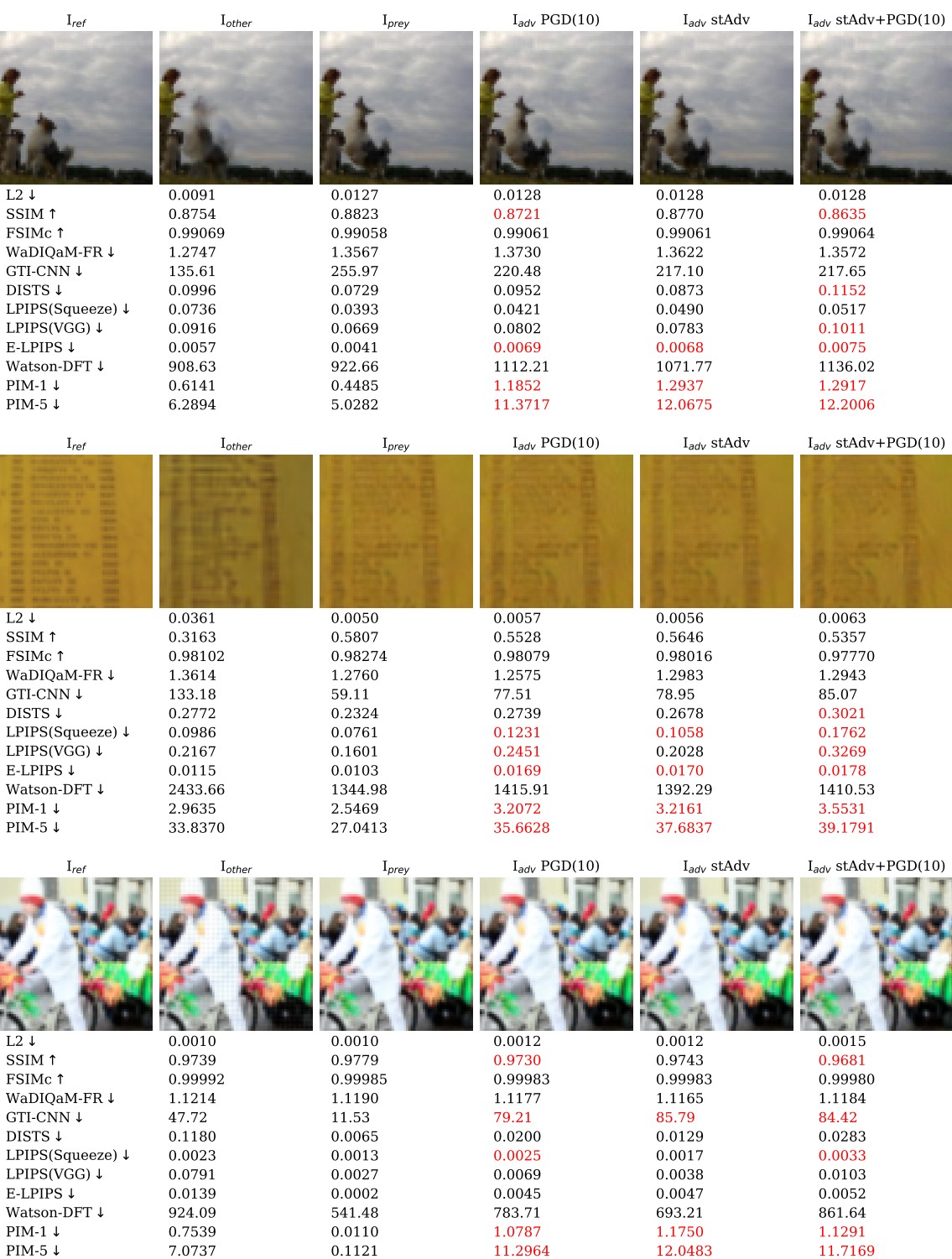

Figure 6: Transferable attack on perceptual similarity metrics. In example 1 (Top), the RMSE between $I_{prey}$ and $I_{adv}$ images (left to right) is 1.26, 2.89, and 2.47. In example 2 (Mid.), the RMSE between $I_{prey}$ and $I_{adv}$ images (left to right) is 1.29, 1.02, and 1.91. In example 3 (Bot.), RMSE between $I_{prey}$ and $I_{adv}$ images (left to right) is 1.43, 1.2, and 2.15. Please refer to Figure 7 in Appendix A for more examples. Text in red indicates that the rank has flipped.

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

# A    Transferable Attack on Perceptual Similarity Metrics

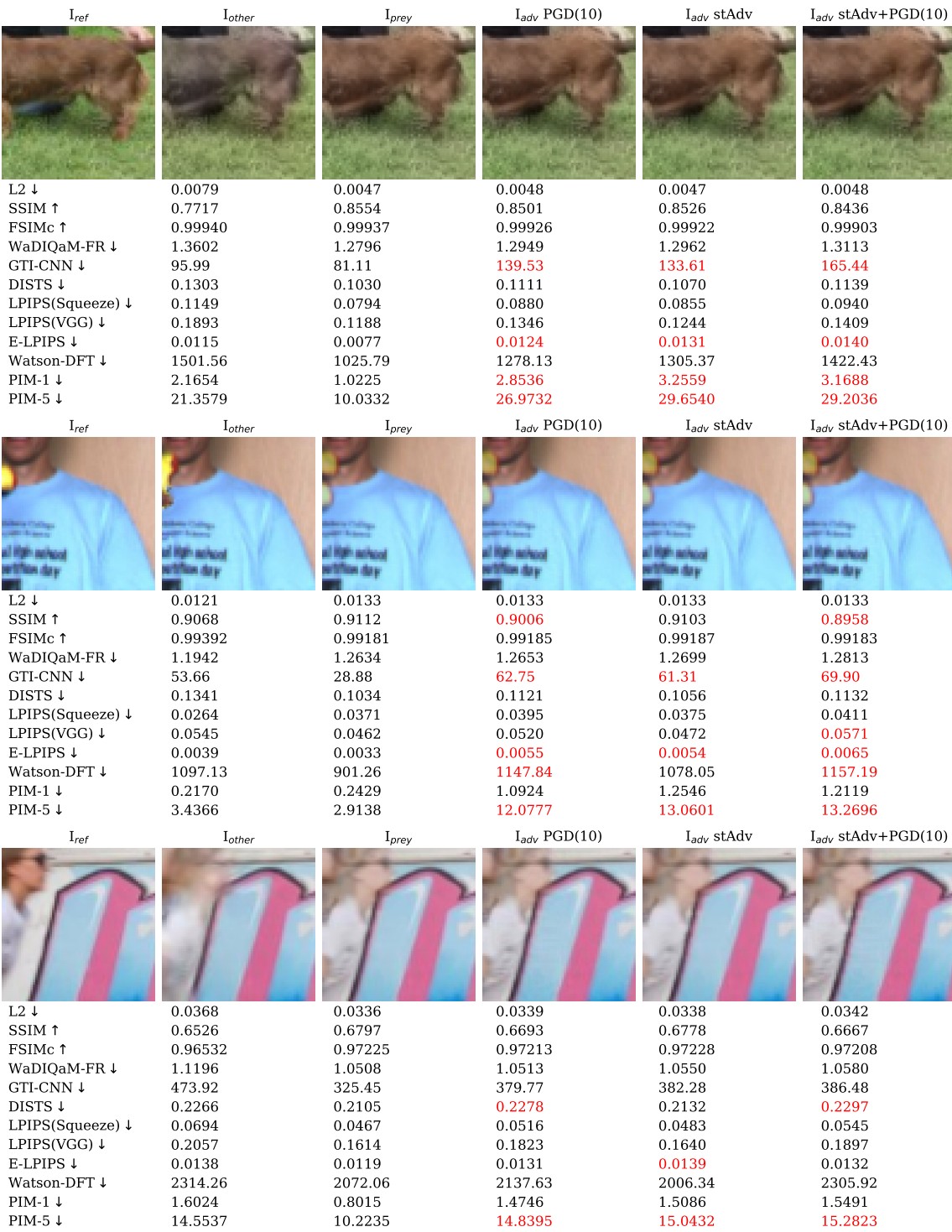

Figure 7: Transferable attack on perceptual similarity metrics. In example 1 (Top), the RMSE between $I_{prey}$ and $I_{adv}$ images (left to right) is 1.35, 1.43, and 2.25. In example 2 (Mid.), the RMSE between $I_{prey}$ and $I_{adv}$ images (left to right) is 1.25, 0.95, and 1.77. In example 3 (Bot.), RMSE between $I_{prey}$ and $I_{adv}$ images (left to right) is 1.37, 0.99, and 2.0. Text in red indicates that the rank has flipped.

## B FGSM Attack on Similarity Metrics

We explain the FGSM attack on perceptual similarity metrics in Algorithm 3.

---

**Algorithm 3:** FGSM attack on Similarity Metrics

---

**Input:** $I_1$, $I_2$, $I_{ref}$, metric $f$, $max\_\epsilon$ (0.05)
**Output:** Least $\epsilon$ value which led to rank flip

**1** $s_0 = f(I_{ref}, I_0)$; $s_1 = f(I_{ref}, I_1)$;
**2** $rank = int(s_0 > s_1)$ // If $I_0$ is more similar to $I_{ref}$ then $rank$ is 0 else 1
**3** **if** $rank = 1$ **then** $I_{prey} = I_1$; $s_{other} = s_0$;
**4** **else** $I_{prey} = I_0$; $s_{other} = s_1$;
**5** $s_{prey} = f(I_{ref}, I_{prey})$
**6** $J = \left((s_{other}/(s_{other} + s_{prey})) - 1\right)^2$ // Loss
**7** $signed\_grad = sign\left(\nabla_{I_{prey}} J\right)$
**8** $\epsilon = 0.0001$
**9** **while** $\epsilon \leq max\_\epsilon$ **do**
**10**   $I_{adv} = I_{prey} + \epsilon \cdot signed\_grad$
**11**   $I_{adv} = clip(I_{adv}, min = -1, max = 1)$ // range [-1,1]
**12**   $s_{adv} = f(I_{ref}, I_{adv})$
**13**   **if** $s_{adv} > s_{other}$ **then**
**14**     **return** *True* // Attack successful
**15**   $\epsilon = \epsilon + 0.0001$
**16** **return** 1 // Largest value of $\epsilon$

---

## C One-Pixel Attack on Similarity Metrics

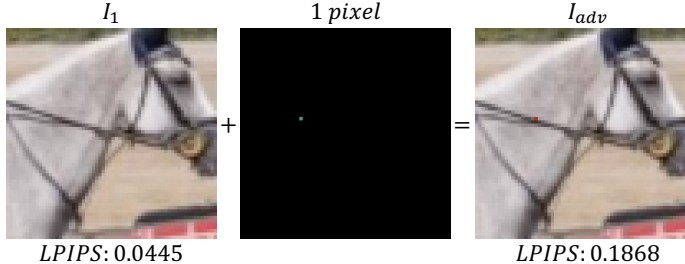

Figure 8: One-pixel attack on LPIPS(Alex). This is a black-box attack as it does not require LPIPS network parameters to generate the adversarial perturbations. The one-pixel perturbation is hardly visible. The RMSE between the prey image $I_1$ and the adversarial image $I_{adv}$ is 1.38.

## D Results on Additional Datasets and High Resolution Images

**Additional datasets.** To test the vulnerability of perceptual similarity on higher image resolutions to adversarial attacks, we use the PieAPP test dataset (Prashnani et al., 2018) and the CLIC validation dataset (CLIC, 2022). The CLIC dataset contains 5220 triplet samples (reference, distorted image A, and distorted image B), and it acts as a test dataset for us since none of the metrics have been trained on it. The PieAPP test set consists of 40 reference images with 15 distorted images per reference image. Out of these, we only select those triplet samples where the preference for distorted image A over B was > 85% and vice versa. Hence, we end up with 1381 samples for our experiment. The original image size of the CLIC samples is 768x768 while for PieAPP is 256x256.

**White-box PGD attack.** We first test the white-box attack on metrics via PGD. As shown in the Tables 6 anf 7, the white-box PGD attack is easily flipping rankings on both datasets. The samples on the PieAPP dataset are harder to flip than the CLIC dataset. We posit that the reason for this lies in the selection criteria for our samples. Since for the PieAPP dataset, we chose only those samples where human preference

Table 6: Whitebox PGD attack results on the PieAPP dataset.

| Network | Image Resolution | Same Rank by Human & Metric | Total Samples | PGD | | | | | |
|---|---|---|---|---|---|---|---|---|---|
| | | | | #Samples Flipped | % pixels with $\epsilon$ | | | RMSE | |
| | | | | | >0.001 | >0.01 | >0.03 | $\mu$ | $\sigma$ |
| L2 | 64x64 | ✓ | 899 | 126/14.0% | 67.5 | 49.9 | 0.0 | 1.7 | 0.8 |
| | | ✗ | 482 | 65/13.5% | 80.2 | 48.6 | 0.0 | 1.7 | 0.9 |
| | 256x256 | ✓ | 963 | 59/6.1% | 87.8 | 69.8 | 0.0 | 2.0 | 0.9 |
| | | ✗ | 418 | 46/11.0% | 85.9 | 62.4 | 0.0 | 2.1 | 1.0 |
| SSIM (Wang et al., 2004) | 64x64 | ✓ | 910 | 391/43.0% | 97.8 | 67.0 | 0.0 | 2.0 | 0.9 |
| | | ✗ | 471 | 120/25.5% | 94.3 | 44.1 | 0.0 | 1.7 | 1.0 |
| | 256x256 | ✓ | 990 | 364/36.8% | 96.7 | 68.9 | 0.0 | 2.1 | 0.9 |
| | | ✗ | 391 | 185/47.3% | 95.0 | 54.2 | 0.0 | 1.8 | 1.0 |
| LPIPS(Alex) (Zhang et al., 2018b) | 64x64 | ✓ | 1016 | 861/84.7% | 90.2 | 30.3 | 0.0 | 1.3 | 0.7 |
| | | ✗ | 365 | 347/95.1% | 89.9 | 31.7 | 0.0 | 1.3 | 0.6 |
| | 256x256 | ✓ | 1184 | 868/73.3% | 90.4 | 39.9 | 0.0 | 1.5 | 0.6 |
| | | ✗ | 197 | 191/97.0% | 84.2 | 20.3 | 0.0 | 1.1 | 0.6 |
| DISTS (Ding et al., 2020) | 64x64 | ✓ | 1041 | 125/12.0% | 97.8 | 73.5 | 0.0 | 2.2 | 0.9 |
| | | ✗ | 340 | 70/20.6% | 96.4 | 65.1 | 0.0 | 2.1 | 1.0 |
| | 256x256 | ✓ | 1286 | 47/3.7% | 97.4 | 73.8 | 0.0 | 2.3 | 1.0 |
| | | ✗ | 95 | 25/26.3% | 95.4 | 70.1 | 0.0 | 2.1 | 1.1 |
| ST-LPIPS(Alex) (Ghildyal & Liu, 2022) | 64x64 | ✓ | 1005 | 823/81.9% | 89.4 | 24.4 | 0.0 | 1.2 | 0.7 |
| | | ✗ | 376 | 370/98.4% | 87.2 | 26.3 | 0.0 | 1.2 | 0.6 |
| | 256x256 | ✓ | 1239 | 599/48.3% | 93.9 | 55.1 | 0.0 | 1.8 | 0.7 |
| | | ✗ | 142 | 138/97.2% | 90.4 | 33.9 | 0.0 | 1.4 | 0.7 |

Table 7: Whitebox PGD attack results on the CLIC dataset.

| Network | Image Resolution | Same Rank by Human & Metric | Total Samples | PGD | | | | | |
|---|---|---|---|---|---|---|---|---|---|
| | | | | #Samples Flipped | % pixels with $\epsilon$ | | | RMSE | |
| | | | | | >0.001 | >0.01 | >0.03 | $\mu$ | $\sigma$ |
| L2 | 256x256 | ✓ | 3167 | 3152/99.5% | 67.6 | 26.6 | 0.0 | 1.1 | 0.6 |
| | | ✗ | 2053 | 2027/98.7% | 67.5 | 19.6 | 0.0 | 1.0 | 0.6 |
| | 512x512 | ✓ | 3120 | 2911/93.3% | 74.8 | 37.2 | 0.0 | 1.4 | 0.8 |
| | | ✗ | 2100 | 1918/91.3% | 74.8 | 30.7 | 0.0 | 1.3 | 0.8 |
| | 768x768 | ✓ | 2992 | 2399/80.2% | 79.8 | 45.4 | 0.0 | 1.6 | 0.9 |
| | | ✗ | 2228 | 1762/79.1% | 80.5 | 48.0 | 0.0 | 1.6 | 0.8 |
| SSIM (Wang et al., 2004) | 256x256 | ✓ | 3307 | 3307/100.0% | 84.2 | 5.7 | 0.0 | 0.8 | 0.4 |
| | | ✗ | 1913 | 1912/99.9% | 76.0 | 6.2 | 0.0 | 0.8 | 0.4 |
| | 512x512 | ✓ | 3200 | 3189/99.7% | 89.1 | 14.6 | 0.0 | 1.0 | 0.5 |
| | | ✗ | 2020 | 2005/99.3% | 85.8 | 11.4 | 0.0 | 0.9 | 0.5 |
| | 768x768 | ✓ | 2997 | 2941/98.1% | 89.9 | 18.6 | 0.0 | 1.1 | 0.6 |
| | | ✗ | 2223 | 2173/97.8% | 87.8 | 14.9 | 0.0 | 1.0 | 0.6 |
| LPIPS(Alex) (Zhang et al., 2018b) | 256x256 | ✓ | 3820 | 3820/100.0% | 54.5 | 0.0 | 0.0 | 0.7 | 0.0 |
| | | ✗ | 1400 | 1400/100.0% | 39.3 | 0.0 | 0.0 | 0.7 | 0.0 |
| | 512x512 | ✓ | 3965 | 3965/100.0% | 64.4 | 0.3 | 0.0 | 0.7 | 0.1 |
| | | ✗ | 1255 | 1255/100.0% | 50.8 | 0.1 | 0.0 | 0.7 | 0.1 |
| | 768x768 | ✓ | 3849 | 3839/99.7% | 73.6 | 2.4 | 0.0 | 0.7 | 0.2 |
| | | ✗ | 1371 | 1371/100.0% | 67.8 | 0.7 | 0.0 | 0.7 | 0.1 |
| DISTS (Ding et al., 2020) | 256x256 | ✓ | 3822 | 3327/87.0% | 97.4 | 55.2 | 0.0 | 1.7 | 0.8 |
| | | ✗ | 1398 | 1308/93.6% | 95.9 | 41.4 | 0.0 | 1.5 | 0.8 |
| | 512x512 | ✓ | 4004 | 2626/65.6% | 98.6 | 72.7 | 0.0 | 2.1 | 0.9 |
| | | ✗ | 1216 | 968/79.6% | 98.2 | 62.4 | 0.0 | 1.9 | 0.9 |
| | 768x768 | ✓ | 3952 | 1286/32.5% | 98.6 | 80.0 | 0.0 | 2.4 | 0.9 |
| | | ✗ | 1268 | 499/39.4% | 96.9 | 69.4 | 0.0 | 2.2 | 0.9 |
| ST-LPIPS(Alex) (Ghildyal & Liu, 2022) | 256x256 | ✓ | 3793 | 3793/100.0% | 56.1 | 0.0 | 0.0 | 0.7 | 0.0 |
| | | ✗ | 1427 | 1427/100.0% | 40.2 | 0.0 | 0.0 | 0.7 | 0.0 |
| | 512x512 | ✓ | 4026 | 4026/100.0% | 70.4 | 0.4 | 0.0 | 0.7 | 0.1 |
| | | ✗ | 1194 | 1194/100.0% | 53.5 | 0.1 | 0.0 | 0.7 | 0.1 |
| | 768x768 | ✓ | 4021 | 4009/99.7% | 81.3 | 5.2 | 0.0 | 0.8 | 0.3 |
| | | ✗ | 1199 | 1199/100.0% | 72.8 | 1.8 | 0.0 | 0.7 | 0.2 |

for a distorted image over the other was > 85%, it seems that the margin between the classes, namely, "less similar" and "more similar" to the reference, is larger, than in the CLIC dataset, making it harder to flip the rank.

**White-box stAdv attack.** We attack the LPIPS(Alex) metric using stAdv on the PieAPP dataset. For the images with higher resolution, it was harder to flip rank. However, that could be due to the settings of our setup. In the loss defined in Equation 4 for the stAdv attack, minimizing $\mathcal{L}_{flow}$ constrains the amount of flow used to generate the adversarial perturbations while minimizing $\mathcal{L}_{rank}$ encourages more perturbations. Hence, if we increase $\alpha$, i.e., the weight for $\mathcal{L}_{rank}$, a larger amount of perturbations would be generated as the flow generating adversarial perturbations will be less constrained. As shown in Table 5, we observe that increasing $\alpha$ helps flipping rank for more samples, but the RMSE of the $I_{adv}$ with $I_{prey}$ is also higher.

Table 8: Whitebox stAdv attack on LPIPS(Alex) on the PieAPP dataset.

| Image Resolution | #Accurate Samples | $\alpha$ from Equation 4 | # Accurate Samples Flipped | RMSE ($\mu/\sigma$) |
|---|---|---|---|---|
| 64x64 | 1016 | 50 | 899/88.5% | 4.3/2.0 |
|  |  | 200 | 1000/98.4% | 5.8/3.1 |
|  |  | 1000 | 1016/100.0% | 7.8/4.9 |
| 256x256 | 1184 | 50 | 28/2.4% | 0.8/0.3 |
|  |  | 200 | 158/13.3% | 2.1/1.3 |
|  |  | 1000 | 566/47.8% | 3.7/1.9 |

**Transferable Adversarial attack.** Here we test the transferable PGD(20) attack. In this experiment, we attack the LPIPS(Alex) metric using the PGD. This experiment is performed on the PieAPP dataset because we found it harder to flip samples on it. Out of the 1184 accurate samples, the rank flipped for 635 samples with a mean RMSE of 1.92 with a standard deviation of 0.15. We test the transferability of these 635 samples to other perceptual similarity metrics. We found that although the metrics did change their scores due to the adversarial perturbations, worsening their prediction, it was still harder to flip ranks on this dataset. Less than 10% of the samples flipped ranks. However, the transferable attack results in Table 9 are consistent with the results on the BAPPS dataset in Table 5 of the main paper. The traditional metrics are more robust than the learned metrics, while the learned metrics are more accurate. The transformer-based metric swinIQA has high accuracy and robustness. E-LPIPS and ST-LPIPS(VGG) which are more robust variants of LPIPS(VGG), showcase more robustness, with ST-LPIPS(VGG) also being more accurate. Similarly, PIM-1 and DISTS are also accurate, along with being more robust. Surprisingly, WaDIQaM-FR showcases higher accuracy on the PieAPP dataset than on the BAPPS dataset, along with being robust on both datasets.

Table 9: Transferable PGD(20) attack on perceptual similarity metrics.

| Network | #Accurate Samples | #Accurate Samples Flipped via PGD(20) |
|---|---|---|
| L2 | 448/71% | 2/0.4% |
| SSIM (Wang et al., 2004) | 456/72% | 17/3.7% |
| MS-SSIM (Wang et al., 2003) | 460/72% | 11/2.4% |
| CWSSIM (Wang & Simoncelli, 2005) | 414/65% | 15/3.6% |
| FSIMc (Zhang et al., 2011) | 461/73% | 4/0.9% |
| WaDIQaM-FR (Bosse et al., 2018) | 602/95% | 13/2.2% |
| GTI-CNN (Ma et al., 2018) | 454/71% | 2/0.4% |
| PieAPP Prashnani et al. (2018) | 476/75% | 8/1.7% |
| LPIPS(Squz.) (Zhang et al., 2018b) | 611/96% | 26/4.3% |
| LPIPS(VGG) (Zhang et al., 2018b) | 554/87% | 63/11.4% |
| E-LPIPS (Kettunen et al., 2019b) | 554/87% | 8/1.4% |
| DISTS (Ding et al., 2020) | 607/96% | 24/4.0% |
| Watson-DFT (Czolbe et al., 2020) | 475/75% | 32/6.7% |
| PIM-1 (Bhardwaj et al., 2020) | 558/88% | 22/3.9% |
| PIM-5 (Bhardwaj et al., 2020) | 550/87% | 33/6.0% |
| A-DISTS (Ding et al., 2021) | 512/81% | 36/7.0% |
| ST-LPIPS(Alex) (Ghildyal & Liu, 2022) | 614/97% | 14/2.3% |
| ST-LPIPS(VGG) (Ghildyal & Liu, 2022) | 584/92% | 25/4.3% |
| SwinIQA (Liu et al., 2022) | 597/94% | 17/2.8% |

# E FGSM versus PGD attack

In our experiments in Table 2, the value chosen for the maximum allowable $\ell_\infty$ perturbation for the PGD attack is lower than that for the FGSM attack. However, if the value is the same, then PGD would be better at flipping the rankings. As shown in Table 10, the PGD attack is more successful than FGSM. In the case of traditional metrics, the results for both attacks are similar. However, for learned perceptual similarity metrics like LPIPS, the number of flips by PGD are greater with a lesser amount of perturbation required.

Table 10: FGSM and PGD attack results when the maximum $\ell_\infty$-norm perturbation is the same for both.

| Network | Same Rank by Human & Metric | Total Samples | FGSM ($\epsilon < 0.03$) | | | | PGD | | | | | |
| | | | #Samples Flipped | Mean $\epsilon$ | RMSE | | #Samples Flipped | % pixels with $\epsilon$ | | | RMSE | |
| | | | | | $\mu$ | $\sigma$ | | >0.001 | >0.01 | >0.03 | $\mu$ | $\sigma$ |
| L2 | ✓ | 9750 | 2419/25% | 0.014 | 1.9 | 1.0 | 2348/24% | 84.4 | 56.1 | 0.0 | 1.9 | 1.0 |
| | ✗ | 2477 | 1220/49% | 0.011 | 1.5 | 1.0 | 1202/49% | 82.0 | 42.7 | 0.0 | 1.5 | 1.0 |
| SSIM | ✓ | 9883 | 5383/54% | 0.012 | 1.7 | 1.0 | 5297/54% | 94.6 | 53.6 | 0.0 | 1.8 | 1.0 |
| (Wang et al., 2004) | ✗ | 2344 | 1851/79% | 0.008 | 1.3 | 0.8 | 1843/79% | 87.3 | 32.0 | 0.0 | 1.3 | 0.8 |
| LPIPS(Alex) | ✓ | 11303 | 5620/50% | 0.012 | 1.7 | 1.0 | 8806/78% | 86.8 | 28.7 | 0.0 | 1.3 | 0.6 |
| (Zhang et al., 2018b) | ✗ | 924 | 897/97% | 0.003 | 0.9 | 0.4 | 917/99% | 59.5 | 3.2 | 0.0 | 0.8 | 0.3 |
| LPIPS(VGG) | ✓ | 10976 | 7431/68% | 0.008 | 1.3 | 0.9 | 9689/88% | 81.6 | 15.6 | 0.0 | 1.0 | 0.5 |
| (Zhang et al., 2018b) | ✗ | 1251 | 1235/99% | 0.002 | 0.8 | 0.4 | 1246/100% | 52.3 | 1.6 | 0.0 | 0.7 | 0.2 |
| DISTS | ✓ | 11158 | 1827/16% | 0.015 | 2.1 | 1.6 | 2306/21% | 97.0 | 75.4 | 0.0 | 2.6 | 1.3 |
| (Ding et al., 2020) | ✗ | 1069 | 643/60% | 0.011 | 1.0 | 1.0 | 723/68% | 91.9 | 50.0 | 0.0 | 2.0 | 1.3 |

In the PGD attack, the step size $\alpha$ is often greater when compared to ours, i.e., 0.001, such that it allows the perturbations to go beyond the maximum $\ell_\infty$-norm threshold $\epsilon$ and then can be projected back to the $\epsilon$ radius. We test with a larger values of $\alpha$, and as shown in Table 11, the severity of the attack increases as $\alpha$ is increased, however, at the expense of more % pixels with perturbation > 0.01. The other parameters for the attack are kept the same.

Table 11: PGD attack results with increasing step size $\alpha$.

| Network | $\alpha$ | Same Rank by Human & Metric | Total Samples | PGD | | | | | |
| | | | | #Samples Flipped | % pixels with $\epsilon$ | | | RMSE | |
| | | | | | >0.001 | >0.01 | >0.03 | $\mu$ | $\sigma$ |
| L2 | 0.00100 | ✓ | 9750 | 2348/24% | 84.4 | 56.1 | 0.0 | 1.9 | 1.0 |
| | | ✗ | 2477 | 1202/49% | 82.0 | 42.7 | 0.0 | 1.5 | 1.0 |
| | 0.00375 | ✓ | 9750 | 2419/25% | 87.3 | 63.8 | 0.0 | 1.9 | 1.0 |
| | | ✗ | 2477 | 1220/49% | 88.2 | 51.0 | 0.0 | 1.6 | 1.0 |
| | 0.00600 | ✓ | 9750 | 2419/25% | 87.3 | 67.9 | 0.0 | 2.2 | 1.1 |
| | | ✗ | 2477 | 1220/49% | 88.2 | 55.6 | 0.0 | 1.8 | 1.1 |
| SSIM | 0.00100 | ✓ | 9883 | 5297/54% | 94.6 | 53.6 | 0.0 | 1.8 | 1.0 |
| | | ✗ | 2344 | 1843/79% | 87.3 | 32.0 | 0.0 | 1.3 | 0.8 |
| | 0.00375 | ✓ | 9883 | 5418/55% | 99.2 | 63.7 | 0.0 | 1.8 | 1.0 |
| | | ✗ | 2344 | 1858/79% | 99.0 | 40.5 | 0.0 | 1.3 | 0.9 |
| | 0.00600 | ✓ | 9883 | 5418/55% | 99.1 | 70.5 | 0.0 | 2.1 | 1.1 |
| | | ✗ | 2344 | 1858/79% | 99.0 | 46.8 | 0.0 | 1.5 | 1.0 |
| LPIPS(Alex) (Zhang et al., 2018b) | 0.00100 | ✓ | 11303 | 8806/78% | 86.8 | 28.7 | 0.0 | 1.3 | 0.6 |
| | | ✗ | 924 | 917/99% | 59.5 | 3.2 | 0.0 | 0.8 | 0.3 |
| | 0.00375 | ✓ | 11303 | 9926/88% | 90.4 | 45.3 | 0.0 | 1.5 | 0.8 |
| | | ✗ | 924 | 920/100% | 93.4 | 7.5 | 0.0 | 0.8 | 0.3 |
| | 0.00600 | ✓ | 11303 | 9994/88% | 88.0 | 55.2 | 0.0 | 1.8 | 0.8 |
| | | ✗ | 924 | 920/100% | 93.4 | 15.1 | 0.0 | 0.9 | 0.4 |
| LPIPS(VGG) (Zhang et al., 2018b) | 0.00100 | ✓ | 10976 | 9689/88% | 81.6 | 15.6 | 0.0 | 1.0 | 0.5 |
| | | ✗ | 1251 | 1246/100% | 52.3 | 1.6 | 0.0 | 0.7 | 0.2 |
| | 0.00375 | ✓ | 10976 | 10322/94% | 89.9 | 29.6 | 0.0 | 1.2 | 0.7 |
| | | ✗ | 1251 | 1248/100% | 95.8 | 3.7 | 0.0 | 0.7 | 0.2 |
| | 0.00600 | ✓ | 10976 | 10337/94% | 88.4 | 40.8 | 0.0 | 1.4 | 0.8 |
| | | ✗ | 1251 | 1248/100% | 96.2 | 7.9 | 0.0 | 0.8 | 0.3 |

## F    Reversing the PGD Attack

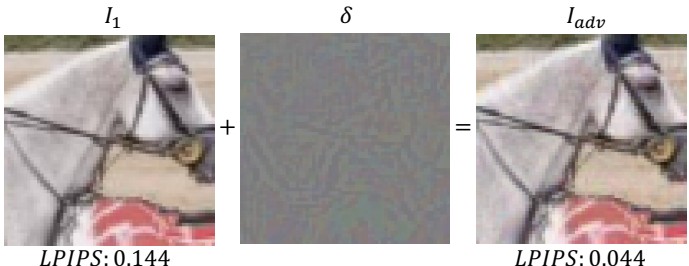

$$I_1 \qquad\qquad \delta \qquad\qquad I_{adv}$$

*LPIPS*: 0.144 $\qquad\qquad\qquad\qquad\qquad$ *LPIPS*: 0.044

Figure 9: PGD attack on LPIPS(Alex). We make the less similar of the two distorted images more similar to the reference image. The RMSE between the prey image $I_0$ and the adversarial image $I_{adv}$ is 4.20.

In this experiment, we try the reverse of the white-box PGD attack in Section 3. For this attack, we do the opposite, i.e., we attack the distorted image that is less similar to $I_{ref}$. Before the attack, the original rank is $s_{other} < s_{prey}$, but after the attack $I_{prey}$ turns into $I_{adv}$, and when the rank flips, $s_{adv} < s_{other}$. We use the LPIPS network parameters to compute the signed gradient via the loss function in Equation 7. As shown in Table 12, it is possible to reverse the attack performed in Table 2.

$$J(\theta, I_{prey}, I_{other}, I_{ref}) = \left( \frac{s_{other}}{s_{other} + s_{prey}} \right)^2 \tag{7}$$

Table 12: Reverse PGD attack results. Here we attack the less similar distorted image and make it more similar to the reference image. Below are the results of the Whitebox PGD attack on the BAPPS dataset.

| Network | Same Rank by Human & Metric | Total Samples | PGD | | | | | |
|---|---|---|---|---|---|---|---|---|
| | | | #Samples Flipped | % pixels with $\epsilon$ | | | RMSE | |
| | | | | >0.001 | >0.01 | >0.03 | $\mu$ | $\sigma$ |
| LPIPS(Alex) | ✓ | 11303 | 6758/59.9% | 87.0 | 27.4 | 0.0 | 1.27 | 0.59 |
| (Zhang et al., 2018b) | ✗ | 924 | 858/92.9% | 60.3 | 5.6 | 0.0 | 0.82 | 0.34 |

