# OpenReview forum: "Attacking Perceptual Similarity Metrics"
_TMLR — Accepted by TMLR_

### Review · Reviewer_RTsU · 2023-03-03

**Summary Of Contributions:**

This paper studies the robustness of various traditional and learned perceptual similarity metrics to imperceptible perturbations. It devises a methodology to craft perturbations via adversarial attacks such as FGSM, PGD, and the One-pixel attack. It finds that the addition of imperceptible distortions can overturn a metric's similarity judgment when comparing two images with a reference. The results indicate that even learned perceptual metrics that match with human similarity judgments are susceptible to such imperceptible adversarial perturbations. It shows that when combined with the PGD attack, the transferability of the adversarial examples can be further increased. It creates a benchmark that provides a good starting point for discussion and further research on the robustness of perceptual similarity metrics to imperceptible adversarial perturbations.

**Audience:**

Yes

**Broader Impact Concerns:**

I don't have any concerns about the ethical implications of the work.

**Claims And Evidence:**

Yes

**Requested Changes:**

1. It should clarify the novelty of the paper more. This would strengthen the work.

2. It should perform experiments on images with high-image resolution. This is critical to securing my recommendation for acceptance.

3. It should clarify the motivation for studying the adversarial robustness of perceptual similarity metrics more. This is critical to securing my recommendation for acceptance.

**Strengths And Weaknesses:**

I think this paper has the following strengths:

1. It studies an interesting and important problem: the robustness of perceptual similarity metrics to imperceptible adversarial perturbations.

2. The paper is well-written and easy to understand. It provides enough background for understanding the paper. The related work is appropriately discussed.

3. The experiments are extensive and can support the main claims of the paper. It also creates a benchmark that provides a good starting point for discussion and further research on the robustness of perceptual similarity metrics to imperceptible adversarial perturbations.

However, I think this paper has the following weaknesses:

1. It just modifies the attack objectives of existing attacks and uses them to attack perceptual similarity metrics. No new attack methods are introduced. Thus, the novelty is limited.

2. In the experiments, the image patches are scaled from size 256$\times$256 to 64$\times$64. Since the low image resolution may affect human judgment and perceptual similarity metrics, it would be good to also show results for larger image resolution.

3. It is unclear whether the adversarial robustness of perceptual similarity metrics is a real concern in practice. It should give some real-world scenarios where the vulnerability of perceptual similarity metrics can cause problems.

---

> ### Author Response · Authors · 2023-03-25
> **Author response to points 1 and 3**
>
> Thank you for your constructive feedback. Below we address the reviewer’s comments:
>
> **1. “It should clarify the novelty of the paper more.”**
>
> - Thank you for this suggestion. While E-LPIPS (Kettunen et al., 2019b) performed a study on the adversarial robustness of LPIPS, a learned perceptual similarity metric, we provide the first systematic study that examines this on a wide variety of perceptual similarity metrics and datasets.
> - Neural networks are susceptible to adversarial attacks; hence the susceptibility of learned perceptual similarity metrics to adversarial attacks is not surprising. However, our study reveals that traditional manually crafted metrics are equally susceptible to such attacks as well, which is surprising. Furthermore, the adversarial examples synthesized for a learned metric could transfer to traditional metrics along with other learned metrics. Some of these metrics have been specifically developed to handle abrupt geometric distortions, but they still lack the ability to handle such transferable adversarial perturbations. Our study will hopefully bring more attention to this problem when considering a perceptual similarity metric for some real-world applications, such as copyright protection of images.
>
> **3. “It is unclear whether the adversarial robustness of perceptual similarity metrics is a real concern in practice. It should give some real-world scenarios where the vulnerability of perceptual similarity metrics can cause problems.”
> “It should clarify the motivation for studying the adversarial robustness of perceptual similarity metrics more.”**
>
> - Perceptual similarity metrics measure the similarity between two images and are widely used in many real-world applications. Having a robust metric is sometimes critical. Copyright protection is one critical use case where automatic image similarity assessment plays an important role. A malicious user can upload copyright-protected images with imperceptible perturbations, making the images less detectable on the internet.

---

> ### Author Response · Authors · 2023-03-25
> **Author response to point 2 (part 1)**
>
> **2. “In the experiments, the image patches are scaled from size 256×256 to 64×64. Since the low image resolution may affect human judgment and perceptual similarity metrics, it would be good to also show results for larger image resolution.”**
>
> - To test the vulnerability of perceptual similarity on higher image resolutions to adversarial attacks, we use the PieAPP test dataset and the CLIC validation dataset. The CLIC dataset contains 5220 triplet samples (reference, distorted image A, and distorted image B), and it acts as a test dataset for us since none of the metrics have been trained on it. The PieAPP test set consists of 40 reference images with 15 distorted images per reference image. Out of these, we only select those triplet samples where the preference for distorted image A over B was > 85% and vice versa. Hence, we end up with 1381 samples for our experiment. The original image size of the CLIC samples is 768x768 while for PieAPP is 256. We have added the experiments below to Appendix D of the paper.
>
> - We first test the whitebox attack on metrics via PGD. As shown in Tables 1 and 2 below, the whitebox PGD attack is easily flipping rankings on both datasets. The samples on the PieAPP dataset are harder to flip than the CLIC dataset. We posit that the reason for this lies in the selection criteria for our samples. Since for the PieAPP dataset, we chose only those samples where human preference for a distorted image over the other was $>$ 85\%, it seems that the margin between the classes, namely, "less similar" and "more similar" to the reference, is larger, than in the CLIC dataset, making it harder to flip the rank.
>
> Table 1. Whitebox PGD attack results on the CLIC dataset.
>
> | Network | Image resolution | H=M$^*$ | Total Samples | #Samples Flipped | \% pixels w/ $\epsilon$ ( >0.001 / >0.01 / >0.05 ) | RMSE ( $\mu$ / $\sigma$ ) |
> |:-----:|:-----:|:-----:|:-----:|:-----:|:-----:|:-----:|
> | L2 | 256x256 | &check; | 3167 | 3152 / 99.5% | 67.6 / 26.6 / 0.0 | 1.1 / 0.6 |
> |  |  | &cross; | 2053 | 2027 / 98.7% | 67.5 / 19.6 / 0.0 | 1.0 / 0.6 |
> |  | 512x512 | &check; | 3120 | 2911 / 93.3% | 74.8 / 37.2 / 0.0 | 1.4 / 0.8 |
> |  |  | &cross; | 2100 | 1918 / 91.3% | 74.8 / 30.7 / 0.0 | 1.3 / 0.8 |
> |  | 768x768 | &check; | 2992 | 2399 / 80.2% | 79.8 / 45.4 / 0.0 | 1.6 / 0.9 |
> |  |  | &cross; | 2228 | 1762 / 79.1% | 80.5 / 48.0 / 0.0 | 1.6 / 0.8 |
> ||
> | SSIM | 256x256 | &check; | 3307 | 3307 / 100.0% | 84.2 / 5.7 / 0.0 | 0.8 / 0.4 |
> |  |  | &cross; | 1913 | 1912 / 99.9% | 76.0 / 6.2 / 0.0 | 0.8 / 0.4 |
> |  | 512x512 | &check; | 3200 | 3189 / 99.7% | 89.1 / 14.6 / 0.0 | 1.0 / 0.5 |
> |  |  | &cross; | 2020 | 2005 / 99.3% | 85.8 / 11.4 / 0.0 | 0.9 / 0.5 |
> |  | 768x768 | &check; | 2997 | 2941 / 98.1% | 89.9 / 18.6 / 0.0 | 1.1 / 0.6 |
> |  |  | &cross; | 2223 | 2173 / 97.8% | 87.8 / 14.9 / 0.0 | 1.0 / 0.6 |
> ||
> | LPIPS(Alex) | 256x256 | &check; | 3820 | 3820 / 100.0% | 54.5 / 0.0 / 0.0 | 0.7 / 0.0 |
> |  |  | &cross; | 1400 | 1400 / 100.0% | 39.3 / 0.0 / 0.0 | 0.7 / 0.0 |
> |  | 512x512 | &check; | 3965 | 3965 / 100.0% | 64.4 / 0.3 / 0.0 | 0.7 / 0.1 |
> |  |  | &cross; | 1255 | 1255 / 100.0% | 50.8 / 0.1 / 0.0 | 0.7 / 0.1 |
> |  | 768x768 | &check; | 3849 | 3839 / 99.7% | 73.6 / 2.4 / 0.0 | 0.7 / 0.2 |
> |  |  | &cross; | 1371 | 1371 / 100.0% | 67.8 / 0.7 / 0.0 | 0.7 / 0.1 |
> ||
> | DISTS | 256x256 | &check; | 3822 | 3327 / 87.0% | 97.4 / 55.2 / 0.0 | 1.7 / 0.8 |
> |  |  | &cross; | 1398 | 1308 / 93.6% | 95.9 / 41.4 / 0.0 | 1.5 / 0.8 |
> |  | 512x512 | &check; | 4004 | 2626 / 65.6% | 98.6 / 72.7 / 0.0 | 2.1 / 0.9 |
> |  |  | &cross; | 1216 | 968 / 79.6% | 98.2 / 62.4 / 0.0 | 1.9 / 0.9 |
> |  | 768x768 | &check; | 3952 | 1286 / 32.5% | 98.6 / 80.0 / 0.0 | 2.4 / 0.9 |
> |  |  | &cross; | 1268 | 499 / 39.4% | 96.9 / 69.4 / 0.0 | 2.2 / 0.9 |
> ||
> | ST-LPIPS(Alex) | 256x256 | &check; | 3793 | 3793 / 100.0% | 56.1 / 0.0 / 0.0 | 0.7 / 0.0 |
> |  |  | &cross; | 1427 | 1427 / 100.0% | 40.2 / 0.0 / 0.0 | 0.7 / 0.0 |
> |  | 512x512 | &check; | 4026 | 4026 / 100.0% | 70.4 / 0.4 / 0.0 | 0.7 / 0.1 |
> |  |  | &cross; | 1194 | 1194 / 100.0% | 53.5 / 0.1 / 0.0 | 0.7 / 0.1 |
> |  | 768x768 | &check; | 4021 | 4009 / 99.7% | 81.3 / 5.2 / 0.0 | 0.8 / 0.3 |
> |  |  | &cross; | 1199 | 1199 / 100.0% | 72.8 / 1.8 / 0.0 | 0.7 / 0.2 |
>
> (*)H=M : Same Rank by Human & Metric
>
> Note: Please see part 2 for Table 2

---

> ### Author Response · Authors · 2023-03-25
> **Author response to point 2 (part 2)**
>
> **Contd.. from (part 1)**
>
> Table 2. Whitebox PGD attack results on the PIEAPP dataset.
>
> | Network | Image resolution | H=M$^*$ | Total Samples | #Samples Flipped | \% pixels w/ $\epsilon$ ( >0.001 / >0.01 / >0.05 ) | RMSE ( $\mu$ / $\sigma$ ) |
> |:-----:|:-----:|:-----:|:-----:|:-----:|:-----:|:-----:|
> | L2 | 64x64 | &check; | 899 | 126 / 14.0% | 67.5 / 49.9 / 0.0 | 1.7 / 0.8 |
> |  |  | &cross; | 482 | 65 / 13.5% | 80.2 / 48.6 / 0.0 | 1.7 / 0.9 |
> |  | 256x256 | &check; | 963 | 59 / 6.1% | 87.8 / 69.8 / 0.0 | 2.0 / 0.9 |
> |  |  | &cross; | 418 | 46 / 11.0% | 85.9 / 62.4 / 0.0 | 2.1 / 1.0 |
> ||
> | SSIM | 64x64 | &check; | 910 | 391 / 43.0% | 97.8 / 67.0 / 0.0 | 2.0 / 0.9 |
> |  |  | &cross; | 471 | 120 / 25.5% | 94.3 / 44.1 / 0.0 | 1.7 / 1.0 |
> |  | 256x256 | &check; | 990 | 364 / 36.8% | 96.7 / 68.9 / 0.0 | 2.1 / 0.9 |
> |  |  | &cross; | 391 | 185 / 47.3% | 95.0 / 54.2 / 0.0 | 1.8 / 1.0 |
> ||
> | LPIPS(Alex) | 64x64 | &check; | 1016 | 861 / 84.7% | 90.2 / 30.3 / 0.0 | 1.3 / 0.7 |
> |  |  | &cross; | 365 | 347 / 95.1% | 89.9 / 31.7 / 0.0 | 1.3 / 0.6 |
> |  | 256x256 | &check; | 1184 | 868 / 73.3% | 90.4 / 39.9 / 0.0 | 1.5 / 0.6 |
> |  |  | &cross; | 197 | 191 / 97.0% | 84.2 / 20.3 / 0.0 | 1.1 / 0.6 |
> ||
> | DISTS | 64x64 | &check; | 1041 | 125 / 12.0% | 97.8 / 73.5 / 0.0 | 2.2 / 0.9 |
> |  |  | &cross; | 340 | 70 / 20.6% | 96.4 / 65.1 / 0.0 | 2.1 / 1.0 |
> |  | 256x256 | &check; | 1286 | 47 / 3.7% | 97.4 / 73.8 / 0.0 | 2.3 / 1.0 |
> |  |  | &cross; | 95 | 25 / 26.3% | 95.4 / 70.1 / 0.0 | 2.1 / 1.1 |
> ||
> | ST-LPIPS(Alex) | 64x64 | &check; | 1005 | 823 / 81.9% | 89.4 / 24.4 / 0.0 | 1.2 / 0.7 |
> |  |  | &cross; | 376 | 370 / 98.4% | 87.2 / 26.3 / 0.0 | 1.2 / 0.6 |
> |  | 256x256 | &check; | 1239 | 599 / 48.3% | 93.9 / 55.1 / 0.0 | 1.8 / 0.7 |
> |  |  | &cross; | 142 | 138 / 97.2% | 90.4 / 33.9 / 0.0 | 1.4 / 0.7 |
>
> (*)H=M : Same Rank by Human & Metric

---

> ### Author Response · Authors · 2023-03-25
> **Author response to point 2 (part 3)**
>
> **Contd.. from (part 2)**
>
> -  Next, we test the transferable PGD(20) attack. In this experiment, we attack the LPIPS(Alex) metric using the PGD. This experiment is performed on the PieAPP dataset because we found it harder to flip samples on it. Out of the 1184 accurate samples, the rank flipped for 635 samples with a mean RMSE of 1.92 with a standard deviation of 0.15. We test the transferability of these 635 samples to other perceptual similarity metrics. We found that although the metrics did change their scores due to the adversarial perturbations, worsening their prediction, it was still harder to flip ranks on this dataset. Less than 10% of the samples flipped ranks. However, the transferable attack results in Table 3 are consistent with the results on the BAPPS dataset in Table 5 of the main paper. The traditional metrics are more robust than the learned metrics, while the learned metrics are more accurate. The transformer-based metric swinIQA has high accuracy and robustness. E-LPIPS and ST-LPIPS(VGG) which are more robust variants of LPIPS(VGG), showcase more robustness, with ST-LPIPS(VGG) also being more accurate. Similarly, PIM-1 and DISTS are also accurate, along with being more robust. Surprisingly, WaDIQaM-FR showcases higher accuracy on the PieAPP dataset than on the BAPPS dataset, along with being robust on both datasets.
>
> Table 3. Transferable PGD attack results on the PieAPP dataset.
>
> | Network | Total Samples / % | #Samples Flipped / % |
> |:-----:|:-----:|:-----:|
> | L2 | 448 / 71% | 2 / 0.4% |
> | SSIM | 456 / 72% | 17 / 3.7% |
> | MS-SSIM | 460 / 72% | 11 / 2.4% |
> | FSIMc | 461 / 73% | 4 / 0.9% |
> | CW-SSIM | 414 / 65% | 15 / 3.6% |
> | WaDIQaM-FR | 602 / 95% | 13 / 2.2% |
> | GTI-CNN | 454 / 71% | 2 / 0.4% |
> | PieAPP | 476 / 75% | 8 / 1.7% |
> | LPIPS(Squz.) | 611 / 96% | 26 / 4.4% |
> | LPIPS(VGG) | 554 / 87% | 63 / 11.4% |
> | E-LPIPS | 554 / 87% | 8 / 1.4% |
> | DISTS | 607 / 96% | 24 / 4.0% |
> | A-DISTS | 512 / 81% | 36 / 7.0% |
> | Watson-DFT | 475 / 75% | 32 / 6.7% |
> | PIM-1 | 558 / 88% | 22 / 3.9% |
> | PIM-5 | 550 / 87% | 33 / 6.0% |
> | ST-LPIPS(Alex) | 614 / 97% | 14 / 2.3% |
> | ST-LPIPS(VGG) | 584 / 92% | 25 / 4.3% |
> | Swin-IQA | 597 / 94% | 17 / 2.8% |

---

> ### Author Response · Authors · 2023-03-25
> **Author response to point 2 (part 4)**
>
> **Contd.. from (part 3)**
>
> - We attack the LPIPS(Alex) metric using stAdv on the PieAPP dataset. For the images with higher resolution, it was harder to flip rank. However, that could be due to the settings of our setup. In the loss defined in Equation 4 for the stAdv attack, minimizing $\mathcal{L}_ {flow}$ constrains the amount of flow used to generate the adversarial perturbations while minimizing $\mathcal{L}_ {rank}$ encourages more perturbations. Hence, if we increase $\alpha$, i.e., the weight for $\mathcal{L}_ {rank}$, a larger amount of perturbations would be generated as the flow generating adversarial perturbations will be less constrained. As shown in Table 4, we observe that increasing $\alpha$ helps flipping rank for more samples, but the RMSE of the $I_{adv}$ with $I_{prey}$ is also higher.
>
> Table 4. Whitebox stAdv attack on LPIPS(Alex) on the PieAPP dataset.
>
> | Image Resolution | # Accurate Samples | $\alpha$ from Equation 4  | # Accurate Samples Flipped / % | RMSE ( $\mu$ / $\sigma$ ) |
> |:-----:|:-----:|:-----:|:-----:|:-----:|
> | 64x64 | 1016 | 50  | 899 / 88.5%  | 4.3 / 2.0 |
> |       |      | 200 | 1000 / 98.4% | 5.8 / 3.1 |
> |       |      | 1000 | 1016 / 100.0% | 7.8 / 4.9 |
> ||
> | 256x256 | 1184 | 50  | 28 / 2.4% | 0.8 / 0.3 |
> |         |      | 200 | 158 / 13.3% | 2.1 / 1.3 |
> |         |      | 1000 | 566 / 47.8% | 3.7 / 1.9 |

---

> > ### Comment · Reviewer_RTsU · 2023-03-27
> > **Thank you for your response**
> >
> > After reading the authors' response, my concerns have been addressed and I have no further questions at this point. I will discuss this with other reviewers to make the final recommendation.

---

### Review · Reviewer_oM18 · 2023-03-08

**Summary Of Contributions:**

This paper studies the robustness of perceptual similarity metrics regarding adversarial attacks. Specifically, the authors craft adversarial perturbations using FGSM, PGD, stAdv, and single-pixel attack and find they can easily flip the rank in two-alternative forced-choice experiments across various perceptual similarity metrics, which indicates the vulnerability of the

**Audience:**

Yes

**Broader Impact Concerns:**

No broader impact statement is present and does not seem necessary.

**Claims And Evidence:**

No

**Requested Changes:**

See Strength and Weakness.

**Strengths And Weaknesses:**

### Strength
1. The topic studied in this paper is interesting. Considering we always use perceptual similarity metrics to help us develop new algorithms in low-level vision tasks, the study on metric robustness may help us reconsider the progress in recent years.
2. The experimental results are notable and convincing, indicating the vulnerability of current perceptual similarity metrics, including the traditional and the DNN-based ones.

### Weakness
**Major:**
1. The motivation in this paper is unclear. Why should we study the robustness of the perceptual metrics? Which problem would a vulnerable perceptual metric cause? The authors are expected to provide a more detailed discussion to convince the readers of this study's values.
2. Why should we use the two-alternative forced-choice experimental design? Actually, we could use the norm-based constraint as used in attack methods to measure the difference between the original image and the perturbed image. There might be something special in the requirements for perceptual similarity metrics, which is unfamiliar to researchers from other fields (e.g., me from adversarial attacks and defense). The authors are expected to explain this in this paper.
3. What is the difference between employing LPIPS to optimize adversarial examples and adversarially attacking LPIPS directly? It confuses me for a long while reading the Introduction. The authors are encouraged to explain their difference clearly.
4. Why do we use different optimization losses between Eqn (1) and Eqn (6)?

**Minor:**

5. In Summary of Sec. 4,  Figures 7 -> Figure 7.

---

> ### Author Response · Authors · 2023-03-25
> **Author response to points 1, 2, and 3**
>
> Thank you for your constructive comments. Below we address the raised issues.
>
> **1. “The motivation in this paper is unclear. Why should we study the robustness of the perceptual metrics? Which problem would a vulnerable perceptual metric cause? The authors are expected to provide a more detailed discussion to convince the readers of this study's values.”**
>
> - Perceptual similarity metrics measure the similarity between two images and are widely used in many real-world applications. Having a robust metric is sometimes critical. Copyright protection is one critical use case where automatic image similarity assessment plays an important role. A malicious user can upload copyright-protected images with imperceptible perturbations, making the images less detectable on the internet.
>
> We have added this discussion in Section 1.
>
> **2. “Why should we use the two-alternative forced-choice experimental design? Actually, we could use the norm-based constraint as used in attack methods to measure the difference between the original image and the perturbed image. There might be something special in the requirements for perceptual similarity metrics, which is unfamiliar to researchers from other fields (e.g., me from adversarial attacks and defense). The authors are expected to explain this in this paper.”**
>
> - Thank you for the suggestion. It is non-trivial to compare metrics based on a norm-based constraint simply because a change of 10% in metric A's score is not equal to a 10% change in metric B's score. But how does one calculate the fooling rate that measures the susceptibility of a perceptual similarity metric? A straightforward method is to compare all metrics against human perceptual judgment. The 2AFC test gathers human judgment on which of the two distorted images is more similar to the reference image. Using this knowledge, we can benchmark various metrics and test whether their accuracy drops or, i.e. if they flip their judgment when attacked. As stated in the paper, to make it a fair challenge, we only use samples where human opinion completely prefers one distorted image over the other.
>
> We have added this discussion to Section 3 of the paper.
>
> **3. “What is the difference between employing LPIPS to optimize adversarial examples and adversarially attacking LPIPS directly? It confuses me for a long while reading the Introduction. The authors are encouraged to explain their difference clearly.”**
>
> - A limitation in early adversarial robustness studies has been the use of $\ell_p$ norms as a distance metric to judge the imperceptibility of synthesized adversarial perturbations. These attack methods optimized for stronger adversarial perturbations while keeping the perturbations within imperceptibility levels via an $\ell_p$ norm. However, as we now know, $\ell_p$ distance metrics are not a good proxy to human perception, and several learned perceptual similarity metrics have been developed to correlate better with human judgment. Laidlaw et al. (2020) proposed neural perceptual threat models (NPTM) and subsequently a defense method that could generalize well against unforeseen adversarial attacks, in which, instead of an $\ell_p$ norm, the severity, or perceptibility of the adversarial perturbations, is bounded by LPIPS, a learned perceptual similarity metric. Hence, they employed LPIPS in their optimization to generate adversarial examples. However, the robustness of the LPIPS and other perceptual similarity metrics is still an unanswered question.
> - In their study, Laidlaw et al. (2020) showed that the image classifier models adversarially trained using $\ell_p$-bounded perturbations could not defend against spatial attacks and had a negligible accuracy of only 4%. Hence, a mismatch between the adversarial attack and defense models leads to such vulnerabilities. Therefore, based on their NPTM, they introduced Perceptual Adversarial Training (PAT), which successfully defended against various adversarial attacks. We posit that more accurate and robust perceptual metrics can lead to stronger defenses against threats. For this, we first need to study "how robust are perceptual similarity metrics against imperceptible adversarial perturbations." In a recent study, Mahloujifar et al. (2020) [1] showed that a better perception model to test the imperceptibility of adversarial perturbations can lead to stronger robustness guarantees for image classifiers.
>
> We have added this explanation in Section 1.
>
>   [1] Saeed Mahloujifar, Chong Xiang, Vikash Sehwag, Sihui Dai, and Prateek Mittal. Robustness from perception. In International Conference on Learning Representations Workshop on Security and Safety in Machine Learning Systems, 2020.

---

> ### Author Response · Authors · 2023-03-25
> **Author response to points 4 and 5**
>
> **4. Why do we use different optimization losses between Eqn (1) and Eqn (6)?**
>
> - Thank you for asking this question. For the case of FGSM and PGD attacks, we first calculate the loss and then move in the opposite direction of the optimization. This calculation is performed at each step of the attack. However, in the case of stAdv, we want to move in the direction of the optimization and minimize the loss in Equation 6 so as to increase the perturbations on the adversarial image.
>
> **5. In Summary of Sec. 4, Figures 7 -> Figure 7.**
>
> - Thank you for the suggestion. We have changed our sentence to “we showcase a few examples in Figure 6 and Figure 7” for better readability.

---

### Review · Reviewer_hxh8 · 2023-03-10

**Summary Of Contributions:**

The paper presents a systematic examination of the robustness of various traditional and learned perceptual similarity metrics to adversarial perturbations. The authors demonstrate that such perturbations can compromise these metrics, which can overturn a metric's similarity judgment. They also propose a methodology to craft such perturbations via adversarial attacks and show that these attacks can be transferable to other metrics. The paper provides a benchmark for the robustness of metrics to imperceptible adversarial perturbations. The contributions of the paper include identifying vulnerabilities of perceptual similarity metrics and highlighting the need for robustness validation in the design and development of newer metrics.

**Audience:**

Yes

**Broader Impact Concerns:**

Based on the information provided in the paper, it does not appear that the authors have explicitly addressed the limitations of their work or provided an ethical statement. It is important for authors to acknowledge the potential limitations of their study and discuss any ethical considerations related to their research, including the potential impact on society or the use of the findings in a harmful manner. This is especially important in work on adversarial attacks.

**Claims And Evidence:**

Yes

**Requested Changes:**

I would appreciate it if the authors could address the points in my weakness section.

1. The authors should clarify the significance of their study in light of previous works on the susceptibility of neural networks to adversarial attacks. This clarification should address the extent to which the findings presented in this paper extend and complement existing knowledge in the field.
2. The authors should revise the claim that "none of the previous investigations have ever considered attacking perceptual similarity metrics" to acknowledge previous work that has investigated the susceptibility of perceptual similarity metrics to adversarial attacks.
3. The authors should consider evaluating their methodology on additional datasets, using additional perceptual similarity techniques, and using stronger adversarial attack techniques such as AutoAttack.


**Strengths And Weaknesses:**

## Strengths
(+) Relevance: The study of adversarial examples on perceptual similarities is an important topic. After I had a look at the literature, I was surprised that a systematic study of this topic had not already been discussed.
(+) Systematic examination: The paper presents a systematic examination of the robustness of various traditional and learned perceptual similarity metrics to imperceptible adversarial perturbations.
(+) Methodology: The authors propose a methodology to craft such perturbations via adversarial attacks and demonstrate that these attacks can be transferable to other metrics.
(+) Benchmark: The paper provides a benchmark for the robustness of metrics to imperceptible adversarial perturbations.

## Weaknesses
(-) Significance: The result that perceptual similarity metrics are susceptible to adversarial attacks is not really surprising. Since for example the LPIPS leverage “standard” trained neural networks to calculate the perceptual distance between two images, which are well known to be susceptible to adversarial attacks it is not surprising that the metric in itself will be susceptible to adversarial examples.
(-) Overclaim: The authors claim that “none of the previous investigations have ever considered attacking perceptual similarity metrics.” However, in E-LPIPS the authors show “that such learned perceptual similarity metrics (LPIPS) are susceptible to adversarial attacks that dramatically contradict human visual similarity judgment.”
(-) Novelty: The authors do not introduce any novel methodology, since existing datasets, metrics, and attack techniques are used.
(-) The evaluation is sparse: Since the authors aim to systematically examine the robustness on perceptual similarity, more attacks and similarity scores should be taken into account. For example, currently, the AutoAttack is considered the strongest adversarial example technique. Additionally, various perceptual similarity techniques have been proposed such as E-LPIPS, Shift-tolerant perceptual similarity metric, etc. A thorough investigation of these metrics would be beneficial for the readers. Especially, the results for which adversarial attacks do not succeed are of interest for the community.

---

> ### Author Response · Authors · 2023-03-25
> **Author response to points 1 and 2**
>
> Thank you for your valuable comments. Below are responses to points 1 and 2:
>
> **1. “The authors should clarify the significance of their study in light of previous works on the susceptibility of neural networks to adversarial attacks. This clarification should address the extent to which the findings presented in this paper extend and complement existing knowledge in the field.”**
>
> - We thank the reviewer for this comment; we agree with the reviewer that the susceptibility of learned perceptual similarity metrics to adversarial attacks is not surprising. However, it is surprising that the traditional manually crafted metrics are equally susceptible to such attacks, and furthermore, the adversarial examples synthesized for a learned metric could transfer to traditional metrics along with other learned metrics. Some of these metrics have been specifically developed to handle abrupt geometric distortions, but they still lack the ability to handle such transferable adversarial perturbations.
>
> - In addition, this paper thoroughly investigated the susceptibility of perceptual similarity metrics to adversarial attacks. This will hopefully bring more attention to this problem when considering a perceptual similarity metric for some real-world applications, such as copyright protection of images.
>
> **2. The authors claim that “none of the previous investigations have ever considered attacking perceptual similarity metrics.” However, in E-LPIPS the authors show “that such learned perceptual similarity metrics (LPIPS) are susceptible to adversarial attacks that dramatically contradict human visual similarity judgment. The authors should revise the claim that "none of the previous investigations have ever considered attacking perceptual similarity metrics" to acknowledge previous work that has investigated the susceptibility of perceptual similarity metrics to adversarial attacks.**
>
> - Thank you for pointing this out to us. We discussed how the attacks we study have similarities with the attack studied in E-LPIPS (Kettunen et al. (2019b)). However, Kettunen et al. (2019b) only studied one metric, i.e. LPIPS, while our study includes a variety of metrics. We agree with you that our claim isn’t exactly fair and hence we have now revised our claim to “none of the previous investigations have ever considered attacking perceptual similarity metrics, except for E-LPIPS (Kettunen et al. (2019b)) which only studies the LPIPS metric.”

---

> ### Author Response · Authors · 2023-03-25
> **Author response to point 3 (part 1)**
>
> Thank you for your valuable comments. Below are responses to point 3:
>
> **3. “The authors should consider evaluating their methodology on additional datasets, using additional perceptual similarity techniques, and using stronger adversarial attack techniques such as AutoAttack.”**
>
> - As suggested by the reviewer, we have added two more traditional metrics, MS-SSIM, and CW-SSIM, and four more learned metrics, A-DISTS, PieAPP, ST-LPIPS, and SwinIQA, to our study. Table 5 of the main paper has been updated accordingly. We have also added a short description for each additional metric in the related work section. We already had E-LPIPS on our benchmark in Table 5 of the main paper.
>
> - We also added tests on additional datasets, including the PieAPP test dataset and the CLIC validation dataset. More details are added to Appendix D.
>
> - We first test the whitebox attack on metrics via PGD. As shown in Tables 1 and 2 below, the whitebox PGD attack is easily flipping rankings on both datasets. The samples on the PieAPP dataset are harder to flip than the CLIC dataset. We posit that the reason for this lies in the selection criteria for our samples. Since for the PieAPP dataset, we chose only those samples where human preference for a distorted image over the other was $>$ 85\%, it seems that the margin between the classes, namely, "less similar" and "more similar" to the reference, is larger, than in the CLIC dataset, making it harder to flip the rank.
>
> Table 1. Whitebox PGD attack results on the CLIC dataset.
>
> | Network | Image resolution | H=M$^*$ | Total Samples | #Samples Flipped | \% pixels w/ $\epsilon$ ( >0.001 / >0.01 / >0.05 ) | RMSE ( $\mu$ / $\sigma$ ) |
> |:----------------------------------------------------:|:-----:|:-----:|:-----:|:-----:|:---------------------------------------:|:-----------------------------------------------:|
> | L2 | 256x256 | &check; | 3167 | 3152 / 99.5% | 67.6 / 26.6 / 0.0 | 1.1 / 0.6 |
> |  |  | &cross; | 2053 | 2027 / 98.7% | 67.5 / 19.6 / 0.0 | 1.0 / 0.6 |
> |  | 512x512 | &check; | 3120 | 2911 / 93.3% | 74.8 / 37.2 / 0.0 | 1.4 / 0.8 |
> |  |  | &cross; | 2100 | 1918 / 91.3% | 74.8 / 30.7 / 0.0 | 1.3 / 0.8 |
> |  | 768x768 | &check; | 2992 | 2399 / 80.2% | 79.8 / 45.4 / 0.0 | 1.6 / 0.9 |
> |  |  | &cross; | 2228 | 1762 / 79.1% | 80.5 / 48.0 / 0.0 | 1.6 / 0.8 |
> ||
> | SSIM | 256x256 | &check; | 3307 | 3307 / 100.0% | 84.2 / 5.7 / 0.0 | 0.8 / 0.4 |
> |  |  | &cross; | 1913 | 1912 / 99.9% | 76.0 / 6.2 / 0.0 | 0.8 / 0.4 |
> |  | 512x512 | &check; | 3200 | 3189 / 99.7% | 89.1 / 14.6 / 0.0 | 1.0 / 0.5 |
> |  |  | &cross; | 2020 | 2005 / 99.3% | 85.8 / 11.4 / 0.0 | 0.9 / 0.5 |
> |  | 768x768 | &check; | 2997 | 2941 / 98.1% | 89.9 / 18.6 / 0.0 | 1.1 / 0.6 |
> |  |  | &cross; | 2223 | 2173 / 97.8% | 87.8 / 14.9 / 0.0 | 1.0 / 0.6 |
> ||
> | LPIPS(Alex) | 256x256 | &check; | 3820 | 3820 / 100.0% | 54.5 / 0.0 / 0.0 | 0.7 / 0.0 |
> |  |  | &cross; | 1400 | 1400 / 100.0% | 39.3 / 0.0 / 0.0 | 0.7 / 0.0 |
> |  | 512x512 | &check; | 3965 | 3965 / 100.0% | 64.4 / 0.3 / 0.0 | 0.7 / 0.1 |
> |  |  | &cross; | 1255 | 1255 / 100.0% | 50.8 / 0.1 / 0.0 | 0.7 / 0.1 |
> |  | 768x768 | &check; | 3849 | 3839 / 99.7% | 73.6 / 2.4 / 0.0 | 0.7 / 0.2 |
> |  |  | &cross; | 1371 | 1371 / 100.0% | 67.8 / 0.7 / 0.0 | 0.7 / 0.1 |
> ||
> | DISTS | 256x256 | &check; | 3822 | 3327 / 87.0% | 97.4 / 55.2 / 0.0 | 1.7 / 0.8 |
> |  |  | &cross; | 1398 | 1308 / 93.6% | 95.9 / 41.4 / 0.0 | 1.5 / 0.8 |
> |  | 512x512 | &check; | 4004 | 2626 / 65.6% | 98.6 / 72.7 / 0.0 | 2.1 / 0.9 |
> |  |  | &cross; | 1216 | 968 / 79.6% | 98.2 / 62.4 / 0.0 | 1.9 / 0.9 |
> |  | 768x768 | &check; | 3952 | 1286 / 32.5% | 98.6 / 80.0 / 0.0 | 2.4 / 0.9 |
> |  |  | &cross; | 1268 | 499 / 39.4% | 96.9 / 69.4 / 0.0 | 2.2 / 0.9 |
> ||
> | ST-LPIPS(Alex) | 256x256 | &check; | 3793 | 3793 / 100.0% | 56.1 / 0.0 / 0.0 | 0.7 / 0.0 |
> |  |  | &cross; | 1427 | 1427 / 100.0% | 40.2 / 0.0 / 0.0 | 0.7 / 0.0 |
> |  | 512x512 | &check; | 4026 | 4026 / 100.0% | 70.4 / 0.4 / 0.0 | 0.7 / 0.1 |
> |  |  | &cross; | 1194 | 1194 / 100.0% | 53.5 / 0.1 / 0.0 | 0.7 / 0.1 |
> |  | 768x768 | &check; | 4021 | 4009 / 99.7% | 81.3 / 5.2 / 0.0 | 0.8 / 0.3 |
> |  |  | &cross; | 1199 | 1199 / 100.0% | 72.8 / 1.8 / 0.0 | 0.7 / 0.2 |
>
> (*)H=M : Same Rank by Human & Metric
>
> Note: Please see part 2 for Table 2

---

> ### Author Response · Authors · 2023-03-25
> **Author response to point 3 (part 2)**
>
> **Contd.. from (part 1)**
>
> Table 2. Whitebox PGD attack results on the PIEAPP dataset.
>
> | Network | Image resolution | H=M$^*$ | Total Samples | #Samples Flipped | \% pixels w/ $\epsilon$ ( >0.001 / >0.01 / >0.05 ) | RMSE ( $\mu$ / $\sigma$ ) |
> |:-----:|:-----:|:-----:|:-----:|:-----:|:-----:|:-----:|
> | L2 | 64x64 | &check; | 899 | 126 / 14.0% | 67.5 / 49.9 / 0.0 | 1.7 / 0.8 |
> |  |  | &cross; | 482 | 65 / 13.5% | 80.2 / 48.6 / 0.0 | 1.7 / 0.9 |
> |  | 256x256 | &check; | 963 | 59 / 6.1% | 87.8 / 69.8 / 0.0 | 2.0 / 0.9 |
> |  |  | &cross; | 418 | 46 / 11.0% | 85.9 / 62.4 / 0.0 | 2.1 / 1.0 |
> ||
> | SSIM | 64x64 | &check; | 910 | 391 / 43.0% | 97.8 / 67.0 / 0.0 | 2.0 / 0.9 |
> |  |  | &cross; | 471 | 120 / 25.5% | 94.3 / 44.1 / 0.0 | 1.7 / 1.0 |
> |  | 256x256 | &check; | 990 | 364 / 36.8% | 96.7 / 68.9 / 0.0 | 2.1 / 0.9 |
> |  |  | &cross; | 391 | 185 / 47.3% | 95.0 / 54.2 / 0.0 | 1.8 / 1.0 |
> ||
> | LPIPS(Alex) | 64x64 | &check; | 1016 | 861 / 84.7% | 90.2 / 30.3 / 0.0 | 1.3 / 0.7 |
> |  |  | &cross; | 365 | 347 / 95.1% | 89.9 / 31.7 / 0.0 | 1.3 / 0.6 |
> |  | 256x256 | &check; | 1184 | 868 / 73.3% | 90.4 / 39.9 / 0.0 | 1.5 / 0.6 |
> |  |  | &cross; | 197 | 191 / 97.0% | 84.2 / 20.3 / 0.0 | 1.1 / 0.6 |
> ||
> | DISTS | 64x64 | &check; | 1041 | 125 / 12.0% | 97.8 / 73.5 / 0.0 | 2.2 / 0.9 |
> |  |  | &cross; | 340 | 70 / 20.6% | 96.4 / 65.1 / 0.0 | 2.1 / 1.0 |
> |  | 256x256 | &check; | 1286 | 47 / 3.7% | 97.4 / 73.8 / 0.0 | 2.3 / 1.0 |
> |  |  | &cross; | 95 | 25 / 26.3% | 95.4 / 70.1 / 0.0 | 2.1 / 1.1 |
> ||
> | ST-LPIPS(Alex) | 64x64 | &check; | 1005 | 823 / 81.9% | 89.4 / 24.4 / 0.0 | 1.2 / 0.7 |
> |  |  | &cross; | 376 | 370 / 98.4% | 87.2 / 26.3 / 0.0 | 1.2 / 0.6 |
> |  | 256x256 | &check; | 1239 | 599 / 48.3% | 93.9 / 55.1 / 0.0 | 1.8 / 0.7 |
> |  |  | &cross; | 142 | 138 / 97.2% | 90.4 / 33.9 / 0.0 | 1.4 / 0.7 |
>
> (*)H=M : Same Rank by Human & Metric

---

> ### Author Response · Authors · 2023-03-25
> **Author response to point 3 (part 3)**
>
> **Contd.. from (part 2)**
>
> -  Next, we test the transferable PGD(20) attack. In this experiment, we attack the LPIPS(Alex) metric using the PGD. This experiment is performed on the PieAPP dataset because we found it harder to flip samples on it. Out of the 1184 accurate samples, the rank flipped for 635 samples with a mean RMSE of 1.92 with a standard deviation of 0.15. We test the transferability of these 635 samples to other perceptual similarity metrics. We found that although the metrics did change their scores due to the adversarial perturbations, worsening their prediction, it was still harder to flip ranks on this dataset. Less than 10% of the samples flipped ranks. However, the transferable attack results in Table 3 are consistent with the results on the BAPPS dataset in Table 5 of the main paper. The traditional metrics are more robust than the learned metrics, while the learned metrics are more accurate. The transformer-based metric swinIQA has high accuracy and robustness. E-LPIPS and ST-LPIPS(VGG) which are more robust variants of LPIPS(VGG), showcase more robustness, with ST-LPIPS(VGG) also being more accurate. Similarly, PIM-1 and DISTS are also accurate, along with being more robust. Surprisingly, WaDIQaM-FR showcases higher accuracy on the PieAPP dataset than on the BAPPS dataset, along with being robust on both datasets.
>
> Table 3. Transferable PGD attack results on the PieAPP dataset.
>
> | Network | Total Samples / % | #Samples Flipped / % |
> |:-----:|:-----:|:-----:|
> | L2 | 448 / 71% | 2 / 0.4% |
> | SSIM | 456 / 72% | 17 / 3.7% |
> | MS-SSIM | 460 / 72% | 11 / 2.4% |
> | FSIMc | 461 / 73% | 4 / 0.9% |
> | CW-SSIM | 414 / 65% | 15 / 3.6% |
> | WaDIQaM-FR | 602 / 95% | 13 / 2.2% |
> | GTI-CNN | 454 / 71% | 2 / 0.4% |
> | PieAPP | 476 / 75% | 8 / 1.7% |
> | LPIPS(Squz.) | 611 / 96% | 26 / 4.4% |
> | LPIPS(VGG) | 554 / 87% | 63 / 11.4% |
> | E-LPIPS | 554 / 87% | 8 / 1.4% |
> | DISTS | 607 / 96% | 24 / 4.0% |
> | A-DISTS | 512 / 81% | 36 / 7.0% |
> | Watson-DFT | 475 / 75% | 32 / 6.7% |
> | PIM-1 | 558 / 88% | 22 / 3.9% |
> | PIM-5 | 550 / 87% | 33 / 6.0% |
> | ST-LPIPS(Alex) | 614 / 97% | 14 / 2.3% |
> | ST-LPIPS(VGG) | 584 / 92% | 25 / 4.3% |
> | Swin-IQA | 597 / 94% | 17 / 2.8% |

---

> ### Author Response · Authors · 2023-03-25
> **Author response to point 3 (part 4)**
>
> **Contd.. from (part 3)**
>
> - We attack the LPIPS(Alex) metric using stAdv on the PieAPP dataset. For the images with higher resolution, it was harder to flip rank. However, that could be due to the settings of our setup. In the loss defined in Equation 4 for the stAdv attack, minimizing $\mathcal{L}_ {flow}$ constrains the amount of flow used to generate the adversarial perturbations while minimizing $\mathcal{L}_ {rank}$ encourages more perturbations. Hence, if we increase $\alpha$, i.e., the weight for $\mathcal{L}_ {rank}$, a larger amount of perturbations would be generated as the flow generating adversarial perturbations will be less constrained. As shown in Table 4, we observe that increasing $\alpha$ helps flipping rank for more samples, but the RMSE of the $I_{adv}$ with $I_{prey}$ is also higher.
>
> Table 4. Whitebox stAdv attack on LPIPS(Alex) on the PieAPP dataset.
>
> | Image Resolution | # Accurate Samples | $\alpha$ from Equation 4  | # Accurate Samples Flipped / % | RMSE ($\mu / \sigma$) |
> |:-----:|:-----:|:-----:|:-----:|:-----:|
> | 64x64 | 1016 | 50  | 899 / 88.5%  | 4.3 / 2.0 |
> |       |      | 200 | 1000 / 98.4% | 5.8 / 3.1 |
> |       |      | 1000 | 1016 / 100.0% | 7.8 / 4.9 |
> ||
> | 256x256 | 1184 | 50  | 28 / 2.4% | 0.8 / 0.3 |
> |         |      | 200 | 158 / 13.3% | 2.1 / 1.3 |
> |         |      | 1000 | 566 / 47.8% | 3.7 / 1.9 |
>
> - The auto attack is an ensemble of four attacks, i.e., Auto-PGD, FAB, and Square attack. It seems that it is readily applicable to the image classification methods, but to adopt it and make it work for perceptual similarity metrics, will require making several changes that don't seem straightforward. Hence, given the time and computing budget, we could not perform this experiment during the rebuttal. We will experiment with it in the future.

---

> ### Author Response · Authors · 2023-03-25
> **Author response to Broader Impact Concerns**
>
> **4. “Based on the information provided in the paper, it does not appear that the authors have explicitly addressed the limitations of their work or provided an ethical statement. It is important for authors to acknowledge the potential limitations of their study and discuss any ethical considerations related to their research, including the potential impact on society or the use of the findings in a harmful manner. This is especially important in work on adversarial attacks.”**
>
> We have added the following Broader Impacts Statement in our paper.
>
> - Perceptual similarity metrics have a wide variety of applications. Hence, there are benefits to studying the robustness of these metrics, and this work presents an opportunity to further improve the alignment of these metrics with human perception. At the same time, it is important to consider the negative outcomes of our work. Exposing the vulnerability of these metrics provides more details to malicious actors who would want to misuse this information to attack applications that make use of these similarity metrics in their pipeline, such as evading copyright detection. Perceptual similarity metrics can also be misused to synthesize malware images that could go undetected online. Therefore, we suggest further research on this topic to include appropriate defenses or more discussion on ways for mitigating such vulnerabilities. To aid further research on this topic, we shall make our code and data publicly available.

---

> > ### Comment · Reviewer_hxh8 · 2023-03-27
> > **Thank you for the author response**
> >
> > I thank the authors for their response. My questions were addressed and I have no further questions at this point. I am looking forward to the discussion with the other reviewers.

---

### Review · Reviewer_EFA2 · 2023-03-28

**Summary Of Contributions:**

The paper studies the adversarial robustness of both traditional and learned metrics for perceptual similarity. In particular, given two distorted images whose similarity to a reference image is computed with a certain metric, it shows that it possible to flip the ranking of the two images via a small adversarial perturbation to one of them. In the experiments, this is shown to happen quite consistently across metrics and threat models for generating the attacks. Finally, the paper studies the transferabitlity across metrics of the adversarial perturbations.

**Audience:**

Yes

**Broader Impact Concerns:**

No concerns.

**Claims And Evidence:**

Yes

**Requested Changes:**

- I think it is necessary to clarify the strengths of the attacks (see above).

**Strengths And Weaknesses:**

Strengths
- Studying the robustness of perceptual metrics is a relevant topic.

- The paper analyzes many existing perceptual metrics, and shows that all are to some extent vulnerable to adversarial perturbations.

Weaknesses
- In Table 2 it seems that FGSM is often more successful than PGD (according to number of samples flipped). This is quite surprising since PGD should lead to a better optimization of the target loss. One reason might be that FGSM is actually tested with several radii, but this would also indicate that PGD is suboptimal: for example, in Alg. 1, with step size $\alpha$ of 0.001 and 40 iterations, the maximal $\ell_\infty$-norm of the perturbation can be 0.04, which is smaller than both the radius in PGD (0.1) and the maximal radius for FGSM (0.05). For PGD the step size and iterations are typically chosen to allow the algorithm to reach the borders of the $\ell_\infty$-ball. Then, it is not clear how effective the attacks used for evaluation are, and if the robustness of the various metrics can be further reduced (this might be important to assess the relative robustness of different metrics).

- As minor point, if I understand it right, in the experiments only the image with initial higher similarity is perturbed: it might be interesting to show also the other way, i.e. increasing the similarity of the other image.

---

> ### Author Response · Authors · 2023-03-30
> **Author response to point 1**
>
> Thank you for your constructive feedback. Below we address the reviewer’s comments:
>
> **1. “In Table 2 it seems that FGSM is often more successful than PGD (according to number of samples flipped). This is quite surprising since PGD should lead to a better optimization of the target loss. … For PGD the step size and iterations are typically chosen to allow the algorithm to reach the borders of the L_inf-ball. Then, it is not clear how effective the attacks used for evaluation are, and if the robustness of the various metrics can be further reduced (this might be important to assess the relative robustness of different metrics).”**
>
> * We apologize for a typo in our Algorithm 1. The maximum number of attack iterations for the PGD attack is 30. This typo has been corrected.
> * Thank you for pointing out that the value chosen for the maximum allowable $\ell_{\infty}$-norm perturbation for the PGD attack is lower than that for the FGSM attack and that if the value is the same, then PGD would be better at flipping the rankings. We agree with the reviewer, and as shown in Table 1 below, the PGD attack is more successful than FGSM. In the case of traditional metrics, the results for both attacks are similar. However, for learned perceptual similarity metrics like LPIPS, the number of flips by PGD are greater with a lesser amount of perturbation required.
> * We had chosen the value of 30 iterations for the PGD attack after visually inspecting for the imperceptibility of perturbations on the generated adversarial samples. Such a low value was chosen to ensure strict imperceptibility. We based the selection of maximum $\epsilon$ for the FGSM attack as 0.05 on empirical evaluation. As shown in Table 2 of the main paper, even with a lesser value for maximum $\ell_{\infty}$ perturbation, the PGD attack outperforms the FGSM attack for learned metrics like LPIPS, DISTS, and WadIQaM-FR.
>
> We have added this discussion to Appendix E and Section 4.1. The new content is marked in blue.
>
> Table 1. FGSM and PGD attack results when the maximum $\ell_{\infty}$-norm perturbation is the same for both.
>
> | Network | H=M$^*$ | Total Samples | $\|$ |  | FGSM ($\epsilon$ < 0.03) |  | $\|$ |  | &nbsp; &nbsp; &nbsp; &nbsp; &nbsp; &nbsp; &nbsp; &nbsp; PGD |  |
> |:-----:|:-----:|:-----:|:-----:|:-----:|:-----:|:-----:|:-----:|:-----:|:-----:|:-----:|
> |  |  |  | $\|$ | **#Samples Flipped** | **Mean $\epsilon$** | **RMSE ( $\mu$ / $\sigma$ )** | $\|$ |**#Samples Flipped** | **\% pixels w/ $\epsilon$ ( >0.001 / >0.01 / >0.03 )** | **RMSE ( $\mu$ / $\sigma$ )** |
> |  |  |  |  |  |  |  |  |  |  |  |  |
> L2  | &check;  | 9750  | $\|$ | 2419 / 25\% |  0.014  | 1.9  / 1.0  | $\|$ | 2348/24\%  | 84.4  / 56.1 / 0.0  | 1.9 / 1.0 |
> | | &cross;  | 2477  | $\|$ | 1220 / 49\%  | 0.011  | 1.5 / 1.0 | $\|$ | 1202/49\%  | 82.0 / 42.7 / 0.0  | 1.5 / 1.0 |
> |  |  |  |  |  |  |  |  |  |  |  |  |
> SSIM  | &check; | 9883  | $\|$ | 5383 / 54\%  | 0.012  | 1.7 / 1.0  | $\|$ | 5297/54\%  | 94.6 / 53.6 / 0.0  | 1.8 / 1.0  |
> | | &cross; | 2344 | $\|$ | 1851 / 79\%  | 0.008  | 1.3 / 0.8  | $\|$ | 1843/79\%  | 87.3 / 32.0 / 0.0  | 1.3 / 0.8 |
> |  |  |  |  |  |  |  |  |  |  |  |  |
> LPIPS(Alex)  | &check; | 11303  |  $\|$ | 5620 / 50\% | 0.012 | 1.7 / 1.0  | $\|$ | 8806 / 78\%  | 86.8 / 28.7 / 0.0  | 1.3 / 0.6 |
> |  | &cross; | 924 | $\|$ | 897 / 97\%  | 0.003  | 0.9 / 0.4 | $\|$ | 917 / 99\% | 59.5 / 3.2 / 0.0 | 0.8 / 0.3 |
> |  |  |  |  |  |  |  |  |  |  |  |  |
> LPIPS(VGG)  | &check;  | 10976 | $\|$ | 7431 / 68\% | 0.008 | 1.3 / 0.9 | $\|$ | 9689 / 88\% | 81.6 / 15.6 / 0.0 | 1.0 / 0.5 |
> |  | &cross; | 1251  | $\|$ | 1235 / 99\%  | 0.002 | 0.8 / 0.4 | $\|$ | 1246 / 100\% | 52.3 / 1.6 / 0.0  | 0.7 / 0.2 |
> |  |  |  |  |  |  |  |  |  |  |  |  |
> DISTS  | &check;  | 11158 | $\|$ | 1827 / 16\% | 0.015 | 2.1 / 1.6 | $\|$ | 2306 / 21\% | 97.0 / 75.4 / 0.0  | 2.6 / 1.3 |
> | | &cross;  | 1069  | $\|$ | 643 / 60\%  | 0.011  | 1.0 / 1.0 | $\|$ | 723 / 68\% | 91.9 / 50.0 / 0.0 | 2.0 / 1.3 |
>
> (*) H=M : Same Rank by Human & Metric

---

> ### Author Response · Authors · 2023-03-30
> **Author response to point 2**
>
> **2. “In the experiments only the image with initial higher similarity is perturbed: it might be interesting to show also the other way, i.e. increasing the similarity of the other image.”**
>
> * We thank the reviewer for this suggestion. We perform this experiment on LPIPS(Alex) and also showcase a visual sample in Appendix F.
> * In this experiment, we try the reverse of the white-box PGD attack in Section 3. For this attack, we do the opposite, i.e., we attack the distorted image that is less similar to $I_{ref}$. Before the attack, the original rank is $s_{other} < s_{prey}$, but after the attack $I_{prey}$ turns into $I_{adv}$, and when the rank flips, $s_{adv} < s_{other}$. We use the LPIPS network parameters to compute the signed gradient via the loss function in the Equation below.
> $$J(\theta, I_{prey}, I_{other}, I_{ref}) = {\bigg( \frac{s_{other}}{s_{other} + s_{prey}} \bigg)}^2$$
>
> * As shown in Table 2 below, it is possible to reverse the attack performed in Table 2 of the main paper.
>
> Table 2. Reverse PGD attack results. Here we attack the less similar distorted image and make it more similar to the reference image. Below are the results of the Whitebox PGD attack on the BAPPS dataset.
> | Network | H=M$^*$ | Total Samples | #Samples Flipped | \% pixels w/ $\epsilon$ ( >0.001 / >0.01 / >0.03 ) | RMSE ( $\mu$ / $\sigma$ ) |
> |:-----:|:-----:|:-----:|:-----:|:-----:|:-----:|
> LPIPS(Alex)  | &check;  | 11303  | 6758 / 59.9\%  | 87.0 / 27.4 / 0.0  | 1.27 / 0.59 |
> | | &cross;  | 924 | 858 / 92.9\% | 60.3 / 5.6 / 0.0 | 0.82 / 0.34 |
>
> (*) H=M : Same Rank by Human & Metric
>
> We have added this discussion to Appendix F. The new content is marked in blue.

---

> > ### Comment · Reviewer_EFA2 · 2023-03-30
> > **Update after rebuttal**
> >
> > I thank the authors for the response and additional experiments.
> >
> > - About the parameters of PGD, I think the current evaluation might still be misleading. In PGD one typically chooses a step size which is larger than the maximal allowed radius $\epsilon$ over number of iterations (something between $\epsilon / 4$ and $\epsilon / 10$ might be a reasonable choice), so that there (likely) is a projection onto the $\ell_\infty$-ball, unlike in the current scheme. In order to limit the perceptibility of the perturbation one can adjust the radius $\epsilon$, while still using the same number of iterations to properly optimize the target loss. I think this would provide a stronger evaluation of the robustness of the metrics (even bounding FGSM to $\epsilon < 0.03$ it is at times better than PGD).
> >
> > - The results of reverse PGD are a nice addition to the evaluation in my opinion.

---

> > > ### Author Response · Authors · 2023-03-31
> > > **Author response to point 3**
> > >
> > > **3. “In PGD one typically chooses a step size which is larger than the maximal allowed radius $\epsilon$ over number of iterations (something between $\epsilon / 4$ and $\epsilon / 10$ might be a reasonable choice), so that there (likely) is a projection onto the $\ell_\infty$-ball, unlike in the current scheme. ... I think this would provide a stronger evaluation of the robustness of the metrics (even bounding FGSM to $\epsilon < 0.03$ it is at times better than PGD).”**
> > >
> > > Thank you for pointing this out to us. We agree with the reviewer that increasing the step size for the PGD attack will increase the number of adversarial perturbations and, subsequently, the flipping of ranks. We test by increasing the step size $\alpha$ from 0.001 to 0.00375 ($\epsilon/8$) and then to 0.006 ($\epsilon/5$). The other parameters for the attack are kept the same. As shown in Table 3 below, the flipping of ranks increases, however, at the expense of more % of pixels with perturbations $>0.01$. We add this discussion in Appendix E and Section 4.1.
> > >
> > > Table 3. PGD attack results with increasing step size $\alpha$
> > >
> > > | Network | Step size $\alpha$ | H=M$^*$ | Total Samples | #Samples Flipped | \% pixels w/ $\epsilon$ ( >0.001 / >0.01 / >0.03 ) | RMSE ( $\mu$ / $\sigma$ ) |
> > > |:-----:|:-----:|:-----:|:-----:|:-----:|:-----:|:-----:|
> > > L2 | 0.00100 | &check; | 9750 | 2348 / 24\% | 84.4 / 56.1 / 0.0 | 1.9 / 1.0 |
> > > |  |  | &cross; | 2477 | 1202 / 49\% | 82.0 / 42.7 / 0.0 | 1.5 / 1.0  |
> > > | | 0.00375 | &check; | 9750 | 2419 / 25\% | 87.3 / 63.8 / 0.0 | 1.9 / 1.0 |
> > > | |  | &cross; | 2477 | 1220 / 49\% | 88.2 / 51.0 / 0.0 | 1.6 / 1.0 |
> > > | | 0.00600 | &check; | 9750 | 2419 / 25\% | 87.3 / 67.9 / 0.0 | 2.2 / 1.1 |
> > > | |  | &cross; | 2477 | 1220 / 49\% | 88.2 / 55.6 / 0.0 | 1.8 / 1.1 |
> > > ||
> > > SSIM | 0.00100 | &check; | 9883 | 5297 / 54\% | 94.6 / 53.6 / 0.0 | 1.8 / 1.0 |
> > > | |  | &cross; | 2344 | 1843 / 79\% | 87.3 / 32.0 / 0.0 | 1.3 / 0.8 |
> > > | | 0.00375 | &check; | 9883 | 5418 / 55\% | 99.2 / 63.7 / 0.0 | 1.8 / 1.0 |
> > > | |  | &cross; | 2344 | 1858 / 79\% | 99.0 / 40.5 / 0.0 | 1.3 / 0.9 |
> > > | | 0.00600 | &check; | 9883 | 5418 / 55\% | 99.1 / 70.5 / 0.0 | 2.1 / 1.1 |
> > > | |  | &cross; | 2344 | 1858 / 79\% | 99.0 / 46.8 / 0.0 | 1.5 / 1.0  |
> > > ||
> > > LPIPS(Alex) | 0.00100 | &check; | 11303 | 8806 / 78\% | 86.8 / 28.7 / 0.0 | 1.3 / 0.6 |
> > > | |  | &cross; | 924 | 917 / 99\% | 59.5 / 3.2 / 0.0 | 0.8 / 0.3 |
> > > | | 0.00375 | &check; | 11303 | 9926 / 88\% | 90.4 / 45.3 / 0.0 | 1.5 / 0.8 |
> > > | |  | &cross; | 924 | 920 / 100\% | 93.4 / 7.5 / 0.0 | 0.8 / 0.3 |
> > > | | 0.00600 | &check; | 11303 | 9994 / 88\% | 88.0 / 55.2 / 0.0 | 1.8 / 0.8 |
> > > | |  | &cross; | 924 | 920 / 100\% | 93.4 / 15.1 / 0.0 | 0.9 / 0.4 |
> > > ||
> > > LPIPS(VGG) | 0.00100 | &check; | 10976 | 9689 / 88\% | 81.6 / 15.6 / 0.0 | 1.0 / 0.5 |
> > > | |  | &cross; | 1251 | 1246 / 100\% | 52.3 / 1.6 / 0.0 | 0.7 / 0.2 |
> > > | | 0.00375 | &check; | 10976 | 10322 / 94\% | 89.9 / 29.6 / 0.0 | 1.2 / 0.7  |
> > > | | | &cross; | 1251 | 1248 / 100\% | 95.8 / 3.7 / 0.0 | 0.7 / 0.2 |
> > > | | 0.00600 | &check; | 10976 | 10337 / 94\% | 88.4 / 40.8 / 0.0 | 1.4 / 0.8 |
> > > | | | &cross; | 1251 | 1248 / 100\% | 96.2 / 7.9 / 0.0 | 0.8 / 0.3 |
> > >
> > > (*) H=M : Same Rank by Human & Metric

---

### Author Response · Authors · 2023-03-25
**Revision of our Paper**

We thank all the reviewers for their constructive comments. We have incorporated their reviews into our revised paper, with the new content marked in blue. We respond to each review individually below.

---

### Decision · Action_Editors · 2023-04-10

**Recommendation:** Accept as is

**Comment:**

This paper studies the reliability of both classical and learned perceptual similarity metrics to adversarial manipulations. Since the evaluation of adversarial attacks heavily depends on the defined metrics, this paper aims to revisit the fundamentals of adversarial examples and highlight the weaknesses of existing metrics. All reviewers unanimously recommend acceptance, and I concur.

I am also recommending Feature Certification based on the criterion that "If this paper was submitted to a top-tier conference, it would likely be presented as an oral/spotlight." My rationale is that as the community of adversarial machine learning is gradually shifting the focus from simple Lp norm perturbations to a more realistic perturbation model, such as the studied perceptual similarity metrics, this paper shows these metrics still have their own flaws. I believe this study will play an important role to push the frontier of research in this field.

**Audience:**

General audience interested in adversarial attacks and defenses.

**Claims And Evidence:**

Yes. The motivation of this paper is well described. The claims are fully supported by the empirical studies.

---

> ### Author Response · Authors · 2023-05-16
> **Author Response**
>
> Thank you for the recommendation. We are grateful to the editor and all reviewers for their valuable feedback. The GitHub link for this paper’s code has been included in the paper and the final camera-ready version has been uploaded.